

# Beyond Fermi's golden rule with the statistical Jacobi approximation

**David M. Long[1,2]⋆, Dominik Hahn[3]†, Marin Bukov[3] and Anushya Chandran[1]**

**1** Department of Physics, Boston University, Boston, Massachusetts 02215, USA
**2** Condensed Matter Theory Center and Joint Quantum Institute, Department of Physics,
University of Maryland, College Park, Maryland 20742, USA
**3** Max Planck Institute for the Physics of Complex Systems, 01187 Dresden, Germany

⋆ dmlong@umd.edu , † hahn@pks.mpg.de

## Abstract

**Many problems in quantum dynamics can be cast as the decay of a single quantum state into a continuum. The time-dependent overlap with the initial state, called the fidelity, characterizes this decay. We derive an analytic expression for the fidelity after a quench to an ergodic Hamiltonian. The expression is valid for both weak and strong quenches, and timescales before finiteness of the Hilbert space limits the fidelity. It reproduces initial quadratic decay and asymptotic exponential decay with a rate which, for strong quenches, differs from Fermi's golden rule. The analysis relies on the *statistical Jacobi approximation* (SJA), which was originally applied in nearly localized systems, and which we here adapt to well-thermalizing systems. Our results demonstrate that the SJA is predictive in disparate regimes of quantum dynamics.**



# 1  Introduction

There are a broad range of problems in quantum dynamics which can be framed as the decay of a single state into a continuum of other states. The archetypal example is the emission or absorption of radiation by an atom [1,2]. However, formally equivalent scenarios arise in the study of thermalization in isolated many-body quantum systems [3,4], heating in driven systems [5], quantum scars [6,7], quantum information [8,9], and Loschmidt echoes used to characterize quantum chaos [9].

The decay of a single state can be captured by the *state fidelity* (also referred to as the *return probability* or *survival probability*),

$$P_0(t) = |\langle \psi_0 | \psi(t) \rangle|^2 \,. \tag{1}$$

We consider the isolated quantum dynamics of a many-body eigenstate $|\psi_0\rangle$ of $H_0$ under the ergodic Hamiltonian $H = H_0 + JV$, such that $|\psi(t)\rangle = e^{-iHt} |\psi_0\rangle$ ($\hbar = 1$). Many analytical methods have already been developed to estimate the fidelity in this context [9–13]. The earliest is a result of first order perturbation theory known as Fermi's golden rule (FGR) [2] (though it was first derived by Dirac [1]). For small $J$ and long times, FGR predicts the exponential decay of $P_0(t)$ with a rate

$$\Gamma_{\text{GR}} = 2\pi J^2 |f_V(0)|^2 \,, \tag{2}$$

where $|f_V(\omega)|^2$ is the spectral function of the perturbation $V$.

In this work, we use the *statistical Jacobi approximation* (SJA) to relax both the requirements of small $J$ and long times. We predict fidelities in well-thermalizing quantum systems for times shorter than a cutoff when finiteness of the Hilbert space limits the fidelity decay, and for perturbation strengths $J$ which are smaller than the total bandwidth $\sigma_E$ of $H_0$. The

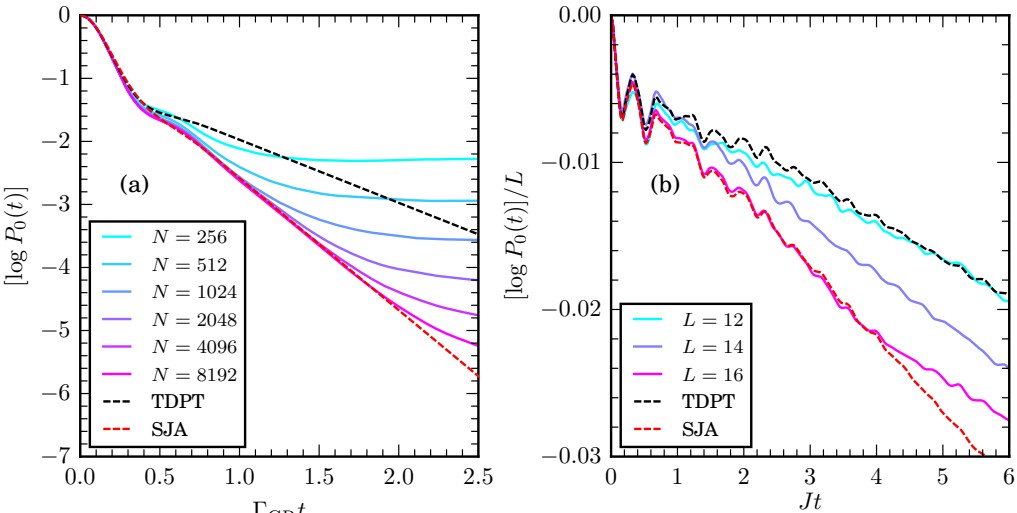

Figure 1: The log-fidelity $P_0(t)$ of an $H_0$ eigenstate decays when the Hamiltonian is quenched to $H = H_0 + JV$. The statistical Jacobi approximation (SJA) correctly predicts all features of the time-dependent fidelity with no free parameters both in (a) random matrix models and (b) local Hamiltonians. The leading order of time-dependent perturbation theory (TDPT, black dashed) predicts exponential decay at the FGR rate $\Gamma_{\text{GR}}$ (2). In both (a) and (b), Eq. (3) (red dashed) correctly predicts a significantly larger decay rate than FGR. *Parameters:* (a) As in the random matrix model of Fig. 8(b). (b) As in the mixed field Ising model of Fig. 10. The TDPT and SJA formulae are calculated using $L = 16$.

perturbation may be large compared to local energy scales. For concision, we denote local energy scales by a single parameter $\sigma_\omega$.

Our main result is a concise integral formula relating the log-fidelity $\log P_0(t)$ to an autocorrelation function of the perturbation,

$$\log P_0(t) = -J^2 \int \mathrm{d}\tau \, (|t - \tau| - |\tau|) \, C_{\text{Jac}}^+(\tau) + \mathcal{O}(J/\sigma_E). \tag{3}$$

Here, $C_{\text{Jac}}^+(\tau) = C_V^+(\tau) + \mathcal{O}(J/\sigma_\omega)^2$ is an autocorrelation function defined through the SJA. At leading order in $J/\sigma_\omega$, it is given by the symmetric connected autocorrelation function of the perturbation in $H_0$, $C_V^+(\tau) = \frac{1}{2} \langle \psi_0 | \{V(\tau), V(0)\} | \psi_0 \rangle_c$. Neglecting higher order contributions to $C_{\text{Jac}}^+$ and replacing it with $C_V^+$ in Eq. (3) correctly reproduces the prediction of conventional time-dependent perturbation theory (TDPT) at leading order in $J/\sigma_\omega$ [14].

Equation (3) has no fit parameters, and accounts for both non-universal early-time dynamics, and the regime of exponential decay—where it accurately predicts corrections to the FGR decay rate (Fig. 1). The formula is derived assuming a continuous density of states, and so does not capture any feature which is due to discreteness of the spectrum.

Numerically, we test Eq. (3) in two classes of models. The first is an ensemble of random matrices, where the spectral function $|f_V(\omega)|^2$ may be chosen arbitrarily, and $\log P_0(t)$ is finite in the limit of infinite Hilbert space dimension, $N \to \infty$ (Fig. 1(a)). Second, we apply Eq. (3) to several well studied spin chains (Fig. 1(b)). In this context, the log-fidelity is extensive (as is the FGR decay rate $\Gamma_{\text{GR}}$), and, for a spin chain of length $L$, it is $[\log P_0(t)]/L$ which has a finite thermodynamic limit. (In $d$ dimensions, $[\log P_0(t)]/L^d$.) Equation (3) shows excellent quantitative agreement with numerics in both cases, outperforming TDPT for some models.

The fidelity has several generic features which are also reproduced by Eq. (3) (Fig. 2). At early times, $\log P_0(t)$ decays quadratically. For late times and weak perturbations, the decay

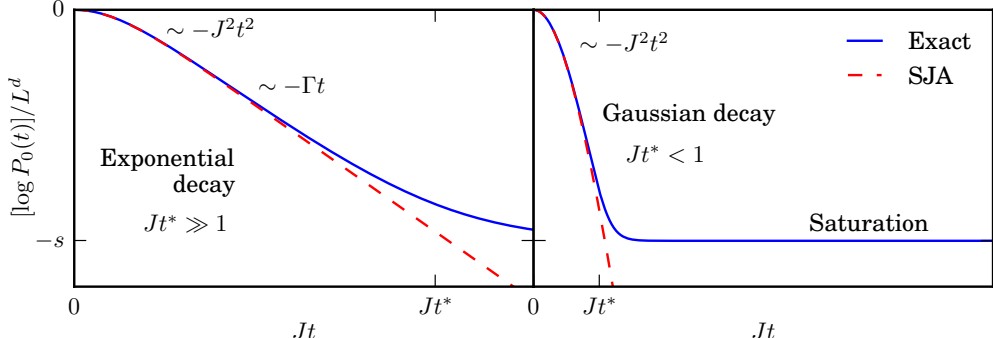

Figure 2: Schematic of the log-fidelity density obtained from Eq. (3) (red dashed) and exact dynamics (blue) in thermalizing $d$-dimensional many-body Hamiltonians. (Left) For weak to intermediate perturbations the log-fidelity initially decays quadratically, $\log P_0(t) \sim -J^2 t^2 L^d$, followed by linear decay, $\log P_0(t) \sim -\Gamma t L^d$ (where $L$ is the linear extent of the system). In the actual dynamics, this decay is cut off at a time $t^*$ when the log-fidelity density reaches the entropy density at the energy of the initial state, $s$. (Right) For strong perturbations, the cutoff time may precede the onset of linear decay, $Jt^* \lesssim 1$. Then decay of the log-fidelity appears quadratic (Gaussian for the fidelity) until the cutoff time.

of $P_0(t)$ is exponential, with a rate predicted by FGR. In any system with a finite Hilbert space, the decay of the fidelity is eventually cut off. In many-body Hamiltonians, the cutoff time can be estimated by equating the log-fidelity density and the entropy density at the energy density of the initial state. The cutoff time $t^*$ is $\mathcal{O}(1)$ with system size, as both the log-fidelity density and the entropy density have finite thermodynamic limits. As such, strong perturbations can push $t^*$ below the onset of exponential decay; $P_0(t)$ then appears Gaussian up to the cutoff time [10] (Fig. 2). The SJA formula Eq. (3) captures fidelity dynamics up to this cutoff time, regardless of whether decay appears exponential or Gaussian.

At the cutoff time $t^*$ and beyond, outside of the regime where Eq. (3) applies, $P_0(t)$ shows a dip below its limit plateau value, and a slow linear growth (or *ramp*) towards that limit. If the density of states contains nonanalyticities due to, for instance, a finite ground state energy, then $P_0(t)$ decays as a power law [10, 15].

Each of these individual features can be predicted by other methods [9–11, 16, 17]. Low orders of time-dependent perturbation theory accurately predict the quadratic decay of $\log P_0(t)$. Indeed, the perturbative series can be cast as an integral transformation of the perturbation's autocorrelation function [9], as in Eq. (3). For later times and weak perturbations, few methods improve qualitatively on FGR [9, 10, 13, 14, 18]. Random matrix techniques can account for late time features [9, 16], including the dip and ramp after the cutoff time (also called a correlation hole) [13, 19, 20]. To our knowledge, previous methods which simultaneously capture all time scales up to the cutoff time only exist for special models, including some random matrix models [12, 13], or weak quenches [14]. In contrast, our method should apply to any ergodic final Hamiltonian, and both weak and strong quenches.

The SJA provides the conceptual and analytical framework for our calculations. This method is a statistical treatment of the Jacobi matrix diagonalization algorithm, which iteratively diagonalizes the Hamiltonian by repeatedly diagonalizing 2×2 submatrices [21, 22]. The method descends from the study of resonances in disordered single-particle systems through strong disorder renormalization groups (SDRG) [23–28], and many-body localization [29–41]. The SJA may be regarded as a kind of SDRG on Hilbert space which diagonalizes the Hamiltonian while maintaining the number of degrees of freedom—similar to Wegner-Wilson

flows [36], though different in the specific RG flow it generates. Reference [42] both framed resonances in terms of the Jacobi algorithm, and extended the method to provide dynamical predictions in prethermal systems. Reference [28] defined a similar method in the context of SDRG in long-range single particle systems, but did not identify a relation to the Jacobi algorithm. This work further extends the SJA to characterize the decay of an initial state into a continuum, which is relevant to ergodic many-body systems. That the same SJA framework may, with distinct approximations, predict dynamics in both prethermal and well-thermalizing systems indicates its broad applicability.

In Sec. 2 we summarize the Jacobi diagonalization algorithm, and related quantities which are central to our analysis. By statistically describing the action of this algorithm on an initial state $|\psi_0\rangle$, we formulate a continuum flow equation for the local density of states (LDOS), the Fourier transform of which gives the fidelity (Sec. 3). We solve the flow equation in Sec. 4 to leading order in the system volume, and assuming that the total density of states (DOS) may be treated as constant. The solution shows excellent agreement with numerics both in random matrix models and more realistic interacting many-body Hamiltonians (Sec. 5). We summarize and discuss applications of our results in Sec. 6.

We defer a technical discussion of the $J/\sigma_E$ dependent corrections in Eq. (3) to Appendix A, where we also demonstrate a connection between the SJA and Dyson Brownian motion [43, 44].

# 2 Statistical Jacobi approximation

The statistical Jacobi approximation (SJA) [42] is the primary analytical tool in this work. This approximation makes a statistical description of the Jacobi diagonalization algorithm to characterize the statistical properties of eigenstates, and hence the dynamics of a quantum system.

In Sec. 2.1 we review the Jacobi diagonalization algorithm, including standard worst-case bounds on its convergence (13). In Sec. 2.2 we identify and characterize two distinct regimes in which the algorithm may operate—a *dense* regime in which the worst-case bounds are close to being saturated, and a *sparse* regime where they are far from being saturated. These two regimes for the algorithm reflect distinct dynamical behaviors of quantum systems.

## 2.1 Jacobi diagonalization algorithm

Originally introduced in 1846 [21], Jacobi's matrix diagonalization algorithm provides a simple iterative procedure to diagonalize an $N \times N$ Hermitian matrix, $H$. While this algorithm is far from the state of the art in modern numerical linear algebra [22], it provides a powerful framework for the description of quantum dynamics [42]. As the diagonalization procedure addresses fast degrees of freedom first, it is possible to relate the action of the algorithm to real time dynamics.

The input for the algorithm is the matrix of $H$ in an arbitrary computational basis $\{|j_0\rangle\}$. The algorithm diagonalizes $H$ using a sequence of $2 \times 2$ rotations which zero—or *decimate*—large off diagonal matrix elements (Fig. 3(a)).

The algorithm proceeds as follows.

1. Identify the largest (in absolute value) off diagonal matrix element of $H$,

$$w_0 = \max_{j_0 \neq k_0} |\langle j_0|H|k_0\rangle| = |\langle a_0|H|b_0\rangle| . \tag{4}$$

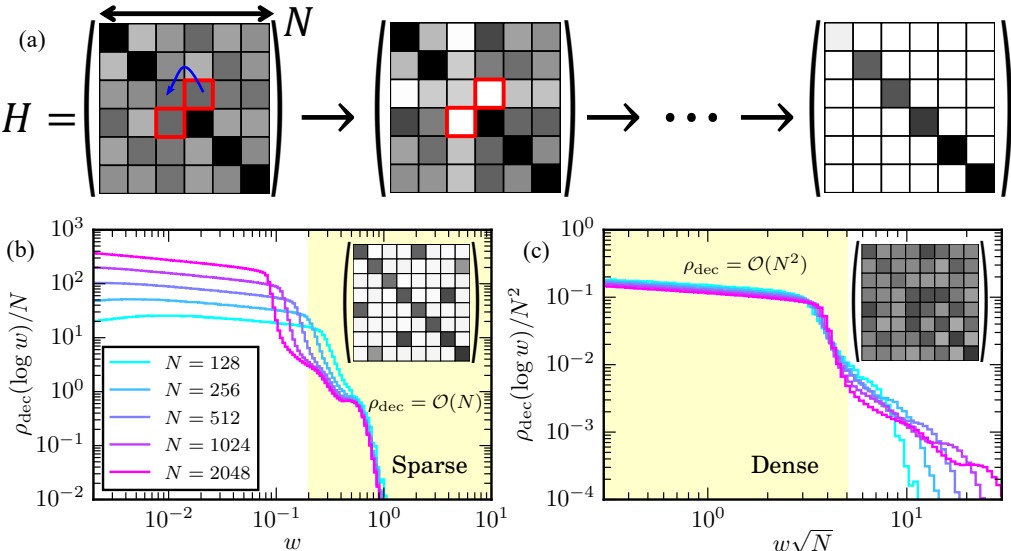

Figure 3: (a) Sketch of the Jacobi diagonalization algorithm. In each iteration, the largest (in absolute value) off diagonal element, $w$, is identified and zeroed (*decimated*) by a $2 \times 2$ rotation. Both (b) and (c) show the same average distribution of $\log w$, $\rho_{\mathrm{dec}}(\log w)$, calculated for the sum of an $N \times N$ random matrix drawn from the Gaussian orthogonal ensemble (GOE) and a random symmetric sparse matrix with 20 nonzero elements per row. (b) For early flow times (larger $w$) the model has a sparse regime [42], where $w = \mathcal{O}(1)$ is much larger than typical elements in its row and column (inset). Consequently, the distribution of decimated elements (15) is $\mathcal{O}(N)$. (c) For late flow times (small $w$) a dense regime emerges, where all matrix elements are of the same scale (inset), $w = \mathcal{O}(N^{-1/2})$. Thus, the distribution of decimated elements becomes $\mathcal{O}(N^2)$. Rescaling by these powers of $N$ produces data collapse in the density of $\log w$.

The $(|a_0\rangle, |b_0\rangle)$ submatrix containing this element is

$$H_{a_0 b_0} = \begin{pmatrix} E_{a_0} & w_0 e^{-i\phi_0} \\ w_0 e^{i\phi_0} & E_{b_0} \end{pmatrix}, \tag{5}$$

where $E_{j_0} = \langle j_0|H|j_0\rangle$, and $\phi_0$ is the complex phase of the off diagonal element.

2. Update the computational basis $\{|j_0\rangle\} \rightarrow \{|j_1\rangle\}$ by applying a $2 \times 2$ rotation between the $|a_0\rangle$ and $|b_0\rangle$ states which diagonalizes the $(|a_0\rangle, |b_0\rangle)$ submatrix. That is, calculating the rotation angle $\eta_0$ by

$$\tan \eta_0 = \frac{2w_0}{E_{a_0} - E_{b_0}}, \tag{6}$$

the update to the basis is

$$|a_0\rangle \rightarrow |a_1\rangle = \cos \tfrac{\eta_0}{2} |a_0\rangle + e^{i\phi_0} \sin \tfrac{\eta_0}{2} |b_0\rangle, \tag{7a}$$

$$|b_0\rangle \rightarrow |b_1\rangle = \cos \tfrac{\eta_0}{2} |b_0\rangle - e^{-i\phi_0} \sin \tfrac{\eta_0}{2} |a_0\rangle, \tag{7b}$$

such that

$$\langle a_1|H|b_1\rangle = 0. \tag{8}$$

Other basis elements are not affected. The full update to the basis is thus

$$|j_0\rangle \rightarrow |j_1\rangle = R_0 |j_0\rangle = \begin{cases} |j_0\rangle \,, & \text{if } j \neq a \text{ and } j \neq b \,, \\ \cos\frac{\eta_0}{2} |a_0\rangle + e^{i\phi_0} \sin\frac{\eta_0}{2} |b_0\rangle \,, & \text{if } j = a \,, \\ \cos\frac{\eta_0}{2} |b_0\rangle - e^{-i\phi_0} \sin\frac{\eta_0}{2} |a_0\rangle \,, & \text{if } j = b \,, \end{cases} \tag{9}$$

where we defined a unitary rotation $R_0$ which implements the update.

3. Finally, return to step 1.

After $n$ iterations, the updated *Jacobi basis* states are

$$|j_n\rangle = R_{n-1} \cdots R_0 |j_0\rangle \,. \tag{10}$$

As the iterations are continued, $n \to \infty$, the basis $\{|j_n\rangle\}$ converges to the eigenbasis of $H$. Indeed, the total norm in the off diagonal of $H$, parameterized as

$$\frac{1}{\beta_n^2} = \frac{1}{N} \sum_{j \neq k} |\langle j_n|H|k_n\rangle|^2 \,, \tag{11}$$

(the right hand side is the mean squared Frobenius norm of a row in the off diagonal) strictly decreases in each iteration

$$\beta_{n+1}^{-2} = \beta_n^{-2} - \frac{2}{N} w_n^2 \,. \tag{12}$$

The element $\langle a_{n+1}|H|b_{n+1}\rangle$ is set to zero, while the sum of squares of other elements in the off diagonal is preserved by Eq. (9). As $w_n$ is the largest off diagonal matrix element, it is at least as large as the root-mean-square, $w_n^2 \geq \beta_n^{-2}/N$. Thus,

$$\beta_{n+1}^{-2} \leq \left(1 - \frac{2}{N^2}\right) \beta_n^{-2} \leq e^{-2/N^2} \beta_n^{-2} \implies \beta_n^{-1} \leq e^{-n/N^2} \beta_0^{-1} \,. \tag{13}$$

The norm of the off diagonal converges exponentially to zero with a rate which is at least $1/N^2$ (Fig. 4). If implemented numerically, this means that the matrix is diagonalized with $\mathcal{O}(N^3)$ floating point operations, similar to other diagonalization algorithms, including the standard QR algorithm.

The elements $w_n$ which are zeroed by the rotations play an important role both in the algorithm, and in our analysis. In the computer science community, these elements are called the *pivotal* elements. In order to emphasize a conceptual link with the renormalization group, we refer to them as *decimated* elements.

Indeed, the algorithm is conceptually similar to the renormalization group (RG): fast degrees of freedom (states with large matrix elements) are updated to eliminate the fast dynamics (zero the matrix element) [28]. However, the Jacobi algorithm differs from the usual formulation of RG as it does not eliminate degrees of freedom. Its action on $H$ is unitary.

The quantity $\beta$ (11) can be viewed as a flow time for the algorithm. In units where $\hbar = 1$, it has units of time, indicating a direct relationship to real time dynamics. When calculating dynamical quantities, such as fidelities, the Jacobi basis at flow time $\beta_n$, $\{|j_n\rangle\}$, defines an approximate Lehmann decomposition,

$$P_0(t) = \left|\langle \psi_0|e^{-iHt}|\psi_0\rangle\right|^2 = \left|\sum_j |\langle j_n|\psi_0\rangle|^2 e^{-iE_{j_n}t}\right|^2 + \mathcal{O}(t/\beta_n)^2 \,. \tag{14}$$

Thus, if a controlled description of the algorithm can be maintained up to a flow time $\beta$, dynamical quantities can be predicted up to a similar timescale.

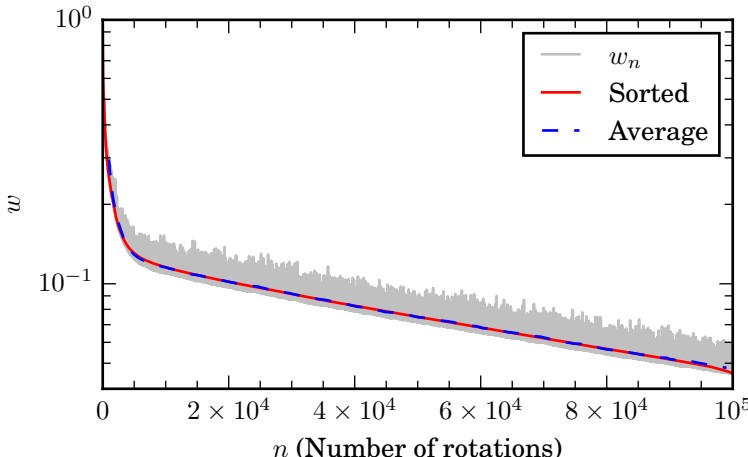

Figure 4: The decimated elements $w_n$ for the model in Fig. 3(b-c), with $N = 1024$. While $w_n$ (grey) does not decrease monotonically with $n$, its typical scale does decrease (exponentially). Either reshuffling decimated elements so that they decrease monotonically (red), or averaging over a fixed number of rotations (the blue dashed curve shows an average over 2000 rotations) causes monotonic decrease by removing small fluctuations.

In certain cases, the operation of the Jacobi algorithm may also admit physical interpretation. In Ref. [42] large rotation angles $\eta$ in the Jacobi algorithm were interpreted as an indicator of resonances between different many-body states. The algorithm also generates dynamics for the energy levels $E_{a_n}$, similar to the Dyson Brownian motion of random matrix theory [43] (Appendix A).

It will be convenient to use the size of the decimated element, $w_n$, to parameterize the flow time. Strictly, this reparameterization requires that $w_n$ is monotonic with $\beta_n$, which is not true. However, the average of $w_n$ over several rotations is a smooth, decreasing function of $\beta_n$ (Fig. 4). Thus, if dealing with average quantities, it is possible to treat $w_n$ as a reparameterization of the flow time.

## 2.2  Distribution of decimated elements

The central object characterizing the statistical properties of the Jacobi algorithm is the *distribution of decimated elements*,

$$\rho_{\text{dec}}(w, E, E') = \sum_n \delta(w - w_n) \big[ \delta(E - E_{a_n}) \delta(E' - E_{b_n}) + \delta(E - E_{b_n}) \delta(E' - E_{a_n}) \big] , \quad (15a)$$

$$\rho_{\text{dec}}(w) = \int dE dE' \, \rho_{\text{dec}}(w, E, E') = 2 \sum_n \delta(w - w_n) . \quad (15b)$$

Here, $|a_n\rangle$ and $|b_n\rangle$ are the states involved in the $n$th rotation, and from the definition $\rho_{\text{dec}}(w, E, E') = \rho_{\text{dec}}(w, E', E)$. If we ignore the effect of the Jacobi rotations updating the elements $\langle j|H|a\rangle$ and $\langle j|H|b\rangle$ (and their transposed partners), then $\rho_{\text{dec}}$ is the joint distribution of Hamiltonian elements $|\langle j|H|k\rangle|$ and their associated energies $E_j$ and $E_k$. However, because the Jacobi algorithm also affects elements other than the one it is decimating (Fig. 3), the distribution $\rho_{\text{dec}}$ differs from the bare distribution of matrix elements. It still satisfies

$$\int_0^\infty dw \, w^2 \rho_{\text{dec}}(w) = N \beta_0^{-2} . \quad (16)$$

That is, the second moment of $\rho_{\text{dec}}(w)$ is the same as the bare distribution of matrix elements. This is because all of the norm of the Hamiltonian in the off diagonal is eventually decimated, and the second moment of $\rho_{\text{dec}}(w)$ is the total decimated norm.

The distribution $\rho_{\text{dec}}$ distinguishes between two qualitatively different regimes of dynamics. In the *sparse regime* (Fig. 3(b)), the largest element in a row of the Hamiltonian, $w$, makes an $\mathcal{O}(1)$ contribution to the total squared Frobenius norm. Decimating one element per row decreases $\beta^{-2}$ by an $\mathcal{O}(1)$ value (11), and, when still in the sparse regime, the new largest element is also smaller by an $\mathcal{O}(1)$ amount. The number of elements decimated in a finite interval of $w$ values thus scales with the Hilbert space dimension,

$$\rho_{\text{dec}}(w, E, E') = \mathcal{O}(N) \quad \text{(sparse)} . \tag{17}$$

However, the worst case bounds on the Jacobi algorithm indicate that it can take as many rotations as there are matrix elements to reduce the norm by an $\mathcal{O}(1)$ factor,

$$\rho_{\text{dec}}(w, E, E') = \mathcal{O}(N^2) \quad \text{(dense)} . \tag{18}$$

This is the *dense regime* (Fig. 3(c)). It emerges when all Hamiltonian matrix elements are of the same scale, and so $w^2 = \mathcal{O}(1/N)$.

Ref. [42] studied the sparse regime in the context of prethermal many-body localization, and found that resonances—large rotation angles $\eta$—accounted for most observable dynamics. Other models which show slower than exponential relaxation—including power-law random banded matrices [45], the Anderson model on the random regular graph [46], and the log normal and Lévy Rosenzweig-Porter models [12, 18, 47]—all possess a sparse regime to which an analysis similar to that in Ref. [42] should apply. (The Rosenzweig-Porter models would require modification of the analysis in Ref. [42], as in those models the ratio of the Frobenius norm of the off-diagonal to the norm of the diagonal may vanish for large system sizes.)

In this work, we focus on the dense regime. As the decimated elements $w$ must decrease as $1/\sqrt{N}$ (more correctly, $1/\sqrt{\nu(E)}$, where $\nu(E)$ is the density of states) the large matrix elements responsible for resonances are rarely encountered. Instead, the dense regime reflects the irreversible decay of fidelities and correlators into a continuum of states. This is the usual scenario at play when using Fermi's golden rule (FGR), and we will find that decay is always asymptotically exponential when the spectral function $|f_V(0)|^2$ is finite. At a technical level, the large number of Jacobi rotations scrambles correlations between Hamiltonian elements, and justifies random-matrix-style approximations, in which we neglect such correlations.

In the context of predicting fidelity decay in the many-body Hamiltonian $H = H_0 + JV$, the initial Jacobi basis $\{|j_0\rangle\}$ is the eigenbasis of $H_0$. If $H_0$ satisfies the eigenstate thermalization hypothesis (ETH) [4, 48–52], then the perturbation $V$ is generically dense in this basis,

$$\langle j_0 | V | k_0 \rangle \sim \frac{f_V(\bar{E}_{j_0 k_0}, \omega_{j_0 k_0})}{\sqrt{\nu(\bar{E}_{j_0 k_0})}} X_{jk} . \tag{19}$$

Here, $|f_V(\bar{E}, \omega)|^2$ is the spectral function for $V$ in the Hamiltonian $H_0$ at mean energy $\bar{E}$ and frequency $\omega$, $\bar{E}_{j_0 k_0} = (E_{j_0} + E_{k_0})/2$, $\omega_{j_0 k_0} = E_{j_0} - E_{k_0}$, $\nu(E)$ is the density of states in $H_0$, and $X_{jk} = \mathcal{O}(1)$ is a random number with variance equal to one. In particular, all matrix elements at a particular energy are of the same scale, implying that the Jacobi algorithm will operate in the dense regime.

If $H_0$ is diagonal in a local basis and the perturbation $V$ has only few-body interactions, then the initial Hamiltonian matrix is sparse. This is the case for (prethermal) MBL [31, 32, 35, 41, 42, 53]. Regardless, the dense regime should emerge at large flow times $\beta$ whenever $H = H_0 + JV$ satisfies ETH. As the Jacobi algorithm comes close to diagonalizing $H$, the Jacobi states present almost as random vectors. The remaining off diagonal of the matrix should then be dense.

# 3 Flow of the local density of states

The Jacobi algorithm, when implemented numerically, diagonalizes the Hamiltonian to within numerical precision. An exact description of all the eigenstates is out of reach analytically, and contains more information than is required to compute fidelities in any case. Rather, only the *distribution* of initial wavefunction amplitudes in the eigenbasis is required. Treating the Jacobi diagonalization algorithm at the level of the statistical distribution is the statistical Jacobi approximation (SJA).

We have

$$P_0(t) = \left| \sum_j |\langle E_j|\psi_0\rangle|^2 e^{-iE_j t} \right|^2 = \left| \int dE\, p(E) e^{-iEt} \right|^2 , \tag{20}$$

where $|E_j\rangle$ is an eigenstate of the full Hamiltonian $H = H_0 + JV$ of energy $E_j$, and

$$p(E) = \frac{1}{dE} \sum_{E_j \in [E,E+dE)} \left| \langle E_j|\psi_0\rangle \right|^2 , \tag{21}$$

has been called the *local density of states* (LDOS) [10], or the *strength function* [3]. Without loss of generality, we assume $V$ to be off diagonal in the eigenbasis of $H_0$, $\{|j_0\rangle\}$.

With some simplifying assumptions (discussed below and summarized in Table 1), it is possible to calculate the LDOS from the decimated distribution $\rho_{dec}$ in the dense regime. In Sec. 3.1, we show that the LDOS in the Jacobi basis,

$$p(w_n, E) = \frac{1}{dE} \sum_{E_{j_n} \in [E,E+dE)} |\langle j_n|\psi_0\rangle|^2 , \tag{22}$$

is determined by its initial condition $p(w_0, E)$ and the flow equation

$$-\partial_w p(w,E) = \int d\omega\, \sin^2 \frac{\eta(\omega)}{2} \tilde{\rho}(w,E,\omega) \left[ p(w,E-\omega)\frac{\nu(E)}{\nu(E-\omega)} - p(w,E) \right] , \tag{23}$$

where

$$\tan\eta(\omega) = \frac{2w}{\omega} , \tag{24}$$

is the rotation angle and we have introduced

$$\tilde{\rho}(w,E,\omega) = \frac{\rho_{dec}(w,E,E-\omega)}{\nu(E)} . \tag{25}$$

Sec. 3.1 contains several technical details, and can be skipped in a first reading of this paper. We find the solution to Eq. (23) in Sec. 4.

## 3.1 Derivation of the flow equation

The action of a single rotation performed by the Jacobi algorithm is easy to characterize. If the Jacobi basis is $\{|j_n\rangle\}$, and the element $w_n = \max_{jk} |\langle j_n|H|k_n\rangle| = |\langle a_n|H|b_n\rangle|$ is to be decimated, then recall that the nontrivial update to the Jacobi basis is (7)

$$|a_{n+1}\rangle = \cos\frac{\eta_n}{2}|a_n\rangle + e^{i\phi_n}\sin\frac{\eta_n}{2}|b_n\rangle , \tag{26a}$$

$$|b_{n+1}\rangle = \cos\frac{\eta_n}{2}|b_n\rangle - e^{-i\phi_n}\sin\frac{\eta_n}{2}|a_n\rangle , \tag{26b}$$

where $\tan\eta_n = 2w_n/(E_{a_n} - E_{b_n})$, and the phase $\phi_n$ will not be important.

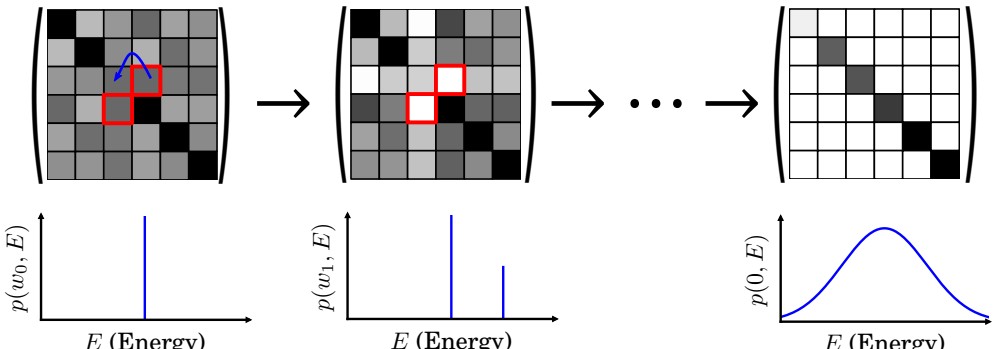

Figure 5: The local density of states (LDOS) $p(w_n, E)$ is the probability density for the initial state $|\psi_0\rangle$ to be in a Jacobi basis state $|j_n\rangle$ with energy $E_{j_n} = E$. The rotations performed by the Jacobi algorithm change the basis $\{|j_n\rangle\}$ (top), and thus the LDOS (bottom), causing a redistribution of probability. The final LDOS, which is related to the fidelity $P_0(t)$ by Fourier transform, is found by running the Jacobi algorithm until $w \to 0$. The approach to this final distribution can be described by a continuum flow equation, Eq. (23).

The corresponding nontrivial update to the probabilities $|\langle j_n|\psi_0\rangle|^2$ is

$$|\langle a_{n+1}|\psi_0\rangle|^2 = \cos^2 \tfrac{\eta_n}{2}|\langle a_n|\psi_0\rangle|^2 + \sin^2 \tfrac{\eta_n}{2}|\langle b_n|\psi_0\rangle|^2$$
$$+ \tfrac{1}{2}\sin\eta_n(e^{-i\phi_n}\langle\psi_0|b_n\rangle\langle a_n|\psi_0\rangle + e^{i\phi_n}\langle\psi_0|a_n\rangle\langle b_n|\psi_0\rangle), \quad (27)$$

which, upon replacing $\cos^2 \tfrac{\eta_n}{2} = 1 - \sin^2 \tfrac{\eta_n}{2}$ and rearranging gives a formula for the change in the probability

$$|\langle a_{n+1}|\psi_0\rangle|^2 - |\langle a_n|\psi_0\rangle|^2 = \sin^2 \tfrac{\eta_n}{2}\left[|\langle b_n|\psi_0\rangle|^2 - |\langle a_n|\psi_0\rangle|^2\right]$$
$$+ \tfrac{1}{2}\sin\eta_n(e^{-i\phi_n}\langle\psi_0|b_n\rangle\langle a_n|\psi_0\rangle + e^{i\phi_n}\langle\psi_0|a_n\rangle\langle b_n|\psi_0\rangle), \quad (28)$$

and similarly for $|\langle b_{n+1}|\psi_0\rangle|^2$.

Making d$n$ Jacobi rotations, the total update to $|\langle j_n|\psi_0\rangle|^2$ is given by the sum over the rotations which affect the state with label $j$. Indexing such rotations by $m$, this is

$$|\langle j_{n+\mathrm{d}n}|\psi_0\rangle|^2 - |\langle j_n|\psi_0\rangle|^2 = \sum_{\substack{m\,:\,n\le m<n+\mathrm{d}n,\\ a_m=j_m}} \sin^2 \frac{\eta_m}{2}\left[|\langle b_m|\psi_0\rangle|^2 - |\langle j_m|\psi_0\rangle|^2\right]$$
$$+ \tfrac{1}{2}\sin\eta_m(e^{-i\phi_m}\langle\psi_0|b_m\rangle\langle j_m|\psi_0\rangle + \text{h.c.}). \quad (29)$$

The expression (29) is too unwieldy for an analytical treatment. It can be made tractable by using $w$ to parameterize flow time, converting the sum on the right hand side to an integral over energy, and replacing the probabilities $|\langle j_m|\psi_0\rangle|^2$ with the probability density $p(w_m, E_j)$. Each of these steps requires that $E_{j_n}$ and $|\langle j_n|\psi_0\rangle|^2$ be slowly varying with $n$ to be valid. Being in the dense regime of Jacobi should imply that this condition is met—many rotations are required to appreciably change the Hamiltonian, so each rotation typically changes little. Thus $E_{j_n}$ and $|\langle j_n|\psi_0\rangle|^2$ will vary slowly.

### 3.1.1 Decimated element as flow time

The decimated elements $w_n$ do not strictly decrease (Fig. 4). Treating $w$ as a continuous parameter controlling the flow time is only possible if $w_n$ is averaged over several rotations. Thus, we must assume that $|\langle j_n|\psi_0\rangle|^2$ and $E_{j_n}$ vary slowly with $n$.

Table 1: The assumptions and approximations made in the technical derivation of the flow equation, Eq. (23), and its solution, Eq. (54). The physical interpretation of each approximation is provided in the second column. All approximations are expected to be valid in the large volume limit of ergodic and local many-body Hamiltonians. The continuum formulation of the LDOS also limits our analysis to times shorter than the cutoff $t^*$.

| Approximation | Interpretation | Text reference |
|---|---|---|
| Dense regime, $\rho_{\text{dec}} = \mathcal{O}(N^2)$ | Assumption of ETH | Sec. 2.2 |
| Continuous, monotonic decrease of the decimated element $w$ | Fluctuations of decimated $w_n$ are small | Fig. 4, Sec. 3.1.1, Eq. (30) |
| Static energy levels $E_{j_n}$ | The quench does not affect the DOS $\nu(E_0)$ | Eq. (33) (relaxed in Appendix A) |
| Continuous LDOS and DOS | Large system size, ETH, and $t < t^*$ | Sec. 3.1.2, Eq. (32), Sec. 3.1.3, Eq. (33), Sec. 4.3.4 |
| Cross terms average to zero | Interference effects are unimportant | Sec. 3.1.3, Eq. (34) |
| DOS much broader than LDOS | Feature of local many-body systems | Sec. 4.1, Eq. (46) (relaxed in Appendix A) |
| Small rotation angle $\eta$ | No resonances | Sec. 3.2.2, Eq. (53) |

Making this assumption, and supposing that the $dn$ rotations reduce the decimated element from $w$ to $w - dw$, the left hand side of Eq. (29) becomes

$$|\langle j_{n+dn}|\psi_0\rangle|^2 - |\langle j_n|\psi_0\rangle|^2 = |\langle j(w-dw)|\psi_0\rangle|^2 - |\langle j(w)|\psi_0\rangle|^2 \to -\partial_w|\langle j(w)|\psi_0\rangle|^2 dw, \quad (30)$$

where we take $dw$ to be infinitesimal, and $|j(w_n)\rangle = |j_n\rangle$.

### 3.1.2 Replacement of the sum by an integral

In replacing the sum with an integral, we must account for the different number of times each pair of states is rotated. This information is encoded in $\rho_{\text{dec}}$. The number of rotations which are made between states of energy $E_{j_m} \in [E, E + dE]$ and $E_{b_m} \in [E', E' + dE']$ while $w_m \in (w - dw, w]$ is

$$\rho_{\text{dec}}(w, E, E') \, dw dE dE'. \quad (31)$$

To find the average number of rotations which affect a *single* state of energy $E_{j_m}$ in the interval $[E, E + dE)$, we divide by the number of states in this shell, $\nu(E)dE$, which gives $\tilde{\rho} \, dw dE'$ as in Eq. (25). The resulting replacement of the sum in Eq. (29) with an integral is

$$\sum_m \to dw \int dE' \, \tilde{\rho}(w, E, E - E'). \quad (32)$$

### 3.1.3 Replacement of probabilities with probability density

Finally, we replace the probabilities $|\langle j(w)|\psi_0\rangle|^2$ with averages over small energy windows,

$$|\langle j(w)|\psi_0\rangle|^2 \to \frac{1}{\nu(E_j)dE} \sum_{E_k \in [E_j, E_j + dE)} |\langle k(w)|\psi_0\rangle|^2 = p(w, E_j)/\nu(E_j). \quad (33)$$

We assume that the cross term in Eq. (29) is negligible compared to the $p(w,E)$ terms,

$$\tfrac{1}{2}\sin\eta_m(e^{-i\phi_m}\langle\psi_0|b_m\rangle\langle j_m|\psi_0\rangle+\text{h.c.})\to 0\,. \tag{34}$$

This term does not have a definite sign, so when neglecting correlations between these terms for different $m$, we expect their average to be a factor $1/\sqrt{\nu(E)dE}$ smaller compared to the $p(w,E)$ terms. Thus, we assume it may be ignored in the limit of infinite system size.

Making all of the substitutions Eqs. (30), (32), (33), and (34), we have

$$-\frac{\partial_w p(w,E)}{\nu(E)}=\int dE'\,\sin^2\frac{\eta(E-E')}{2}\tilde{\rho}(w,E,E-E')\left[\frac{p(w,E')}{\nu(E')}-\frac{p(w,E)}{\nu(E)}\right]. \tag{35}$$

Multiplying both sides by $\nu(E)$ and changing variables to $\omega=E-E'$ in the integral recovers Eq. (23).

Additionally, the energies $E_{j_n}$ are affected by the perturbation, and are altered by the Jacobi evolution. This effect will not be important in the limit we consider in the main text. We leave a derivation and discussion of this addition to the flow equation to Appendix A.

## 3.2 Basic properties

The flow equation Eq. (23) is a linear integro-differential equation. It possesses several expected properties, which we briefly list in this section.

### 3.2.1 Conservation of probability

The flow equation conserves total probability. The rate of change of the total probability is

$$-\partial_w\int dE\,p(w,E)=\int dEdE'\,\sin^2\frac{\eta(E-E')}{2}\tilde{\rho}(w,E,E-E')\left[p(w,E')\frac{\nu(E)}{\nu(E')}-p(w,E)\right]. \tag{36}$$

Exchanging integration variables $E\leftrightarrow E'$ in the right hand side does not change the value of the integral, but multiplies the integrand by $-1$. This implies that the rate of change of the total probability is zero.

Indeed, we have $\sin^2\frac{\eta(E-E')}{2}=\sin^2\frac{\eta(E'-E)}{2}$, while

$$\rho_{\text{dec}}(w,E,E')=\rho_{\text{dec}}(w,E',E)\implies\nu(E)\tilde{\rho}(w,E,E-E')=\nu(E')\tilde{\rho}(w,E',E'-E), \tag{37}$$

so that new form of the integral after exchanging variables is

$$\int dEdE'\,\sin^2\frac{\eta(E-E')}{2}\tilde{\rho}(w,E,E-E')\left[p(W,E')\frac{\nu(E)}{\nu(E')}-p(W,E)\right]$$

$$=\int dE'dE\,\sin^2\frac{\eta(E'-E)}{2}\tilde{\rho}(w,E',E'-E)\left[p(w,E)\frac{\nu(E')}{\nu(E)}-p(w,E')\right]$$

$$=\int dE'dE\,\sin^2\frac{\eta(E-E')}{2}\tilde{\rho}(w,E,E-E')\left[p(w,E)-p(w,E')\frac{\nu(E)}{\nu(E')}\right]. \tag{38}$$

This shows

$$-\partial_w\int dE\,p(w,E)=\partial_w\int dE\,p(w,E)\,, \tag{39}$$

as claimed. Thus

$$-\partial_w\int dE\,p(w,E)=0\,, \tag{40}$$

and the total probability is conserved.

### 3.2.2 Large volume limit

In neglecting the cross-term in the microscopic flow equation Eq. (29) we have already specialized to the limit of large system volume. (More correctly, the limit of large DOS.) In this limit, and in the dense regime, the maximum matrix element $w_0$ should decrease exponentially with the volume. Indeed, for a spatially local $H_0$ which satisfies the ETH, we have

$$w_0^2 = J^2 \mathcal{O}\big(e^{-S(E)}\big), \tag{41}$$

for some energy $E$, and where $S(E)$ is the (extensive) entropy at energy $E$.

The flow equation can be simplified in this limit of small $w_0^2$. From

$$\tan \eta = \frac{2w}{\omega}, \tag{42}$$

we have the small angle expansion

$$\sin^2 \frac{\eta}{2} = \frac{1}{2}\left(1 - \frac{1}{\sqrt{1 + (2w/\omega)^2}}\right) = \frac{w^2}{\omega^2} + \mathcal{O}\big(w^4/\omega^4\big). \tag{43}$$

For the flow equation, this gives

$$-\partial_w p(w, E) \sim \int d\omega \, \frac{w^2}{\omega^2} \tilde{\rho}(w, E, \omega)\left[p(w, E - \omega)\frac{\nu(E)}{\nu(E - \omega)} - p(w, E)\right]. \tag{44}$$

The combination $w^2 \tilde{\rho}$ has a finite integral (Sec. 4), so the right hand side makes a non-zero contribution to the final value of the LDOS, $p(0, E)$.

The flow equation predicts a leading order correction to the LDOS which scales as $w^2/\omega^2$. This structure should be familiar from the first order of perturbation theory in $J$, which is being recovered in this limit of the flow equation.

Note that, in making the small angle approximation for $\eta$, we have neglected any resonances. That is, large rotation angles as a result of an atypically small value of the energy difference $\omega$. Such rotations are always present for any $w_0$, but they make up a vanishing fraction of all rotations, and so may be neglected in the $w_0 \to 0$ limit.

### 3.2.3 Uniform fixed point

The flow equation (23) has a fixed point given by

$$p_{\text{unif}}(E) = \nu(E)/N. \tag{45}$$

This corresponds to the wavefunction $\langle j_n | \psi_0 \rangle$ having equal probability in every Jacobi basis state, so that $|\psi_0\rangle$ appears as a Haar random vector.

However, this fixed point lies outside the regime of applicability of the flow equation for an initially peaked LDOS. When the LDOS flows to a width comparable to the many-body DOS, we can no longer neglect the effect of the perturbation on the DOS. Further, any microcanonical state of a local Hamiltonian $H_0$ has a vanishing energy density variance in $H$, making this fixed point unphysical. As such, the existence of this fixed point solution will not be relevant to our analysis.

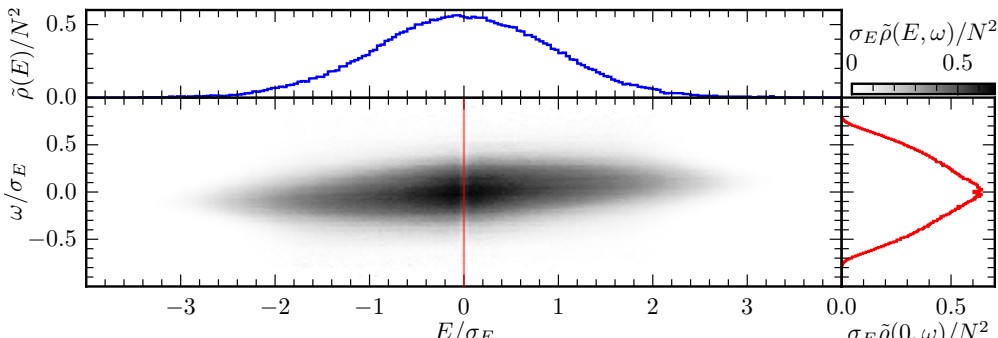

Figure 6: The solution to the flow equation leading to Eq. (3) assumes that $\tilde{\rho}(w, E, \omega)$ (25) is much broader in $E$ than in $\omega$. This feature is visible in (center) a two-dimensional plot of the marginal $\tilde{\rho}(E, \omega) = \int dw \, \tilde{\rho}(w, E, \omega)$, or through comparing (top) the $E$-marginal $\tilde{\rho}(E) = \int d\omega \, \tilde{\rho}(E, \omega)$ with (right) a constant-$E$ cut (shown in red in the center panel) of $\tilde{\rho}(E, \omega)$. Note that the scale of the $E$ and $\omega$ axes are equal, and that $\tilde{\rho}(E)$ is dimensionless, while $\tilde{\rho}(E, \omega)$ is an energy density. *Parameters*: $N = 1024$, $\nu(E) = (N/\sqrt{2\pi})e^{-E^2/2\sigma_E^2}$, $|f_V(E, \omega)|^2 = (2\sigma_\omega)^{-1}\cosh^{-2}(E/\sigma_E)\cosh^{-2}(\omega/\sigma_\omega)$, and $\sigma_\omega/\sigma_E = J/\sigma_E = 0.1$ in the random matrix model of Eq. (77).

## 4 Solution of the flow equation

### 4.1 Solution for broad density of states

The flow equation for the LDOS is difficult to solve in closed form for arbitrary initial LDOS. However, upon specializing to narrow LDOS (discussed below), it can be solved exactly. We leave a discussion of corrections for broad LDOS to Appendix A.

Identifying a characteristic width in $E$, $\sigma_E$, for $\tilde{\rho}(w, E, \omega)$, we solve the flow equation at the leading order in $\sigma_E^{-1}$ (which is $\sigma_E^0$). The solution holds when the magnitude of the perturbation ($J$), any characteristic width of $\tilde{\rho}(w, E, \omega)$ in $\omega$ ($\sigma_\omega$), and the initial width of the LDOS are all much smaller than $\sigma_E$ (Fig. 6). In many-body Hamiltonians, this is usually the case. The total energy $E$ is an extensive variable, so the variation of $\tilde{\rho}(w, E, \omega)$—which we will see is related to a spectral function of the perturbation—in $E$ is suppressed compared to its variation in the energy difference $\omega$. In random matrix models, this regime must be engineered as part of the model definition. (Outside of particular models, the widths $\sigma_{\omega,E}$ will be used at the level of dimensional analysis, and so we do not define them precisely.)

At leading order in the ratios $\sigma_\omega/\sigma_E$ and $J/\sigma_E$, we may make the replacements

$$\frac{\nu(E)}{\nu(E-\omega)} \approx 1, \quad \text{and} \quad \tilde{\rho}(w, E, \omega) \approx \tilde{\rho}(w, E_0, \omega), \tag{46}$$

where $p(w_0, E)$ is peaked at $E = E_0$. This assumption is recovered in a more systematic expansion in derivatives of $\tilde{\rho}(w, E, \omega)$ with respect to $E$ in Appendix A.

With this assumption, the flow equation becomes

$$-\partial_w p(w, E) = \int d\omega \, \sin^2 \frac{\eta(\omega)}{2} \tilde{\rho}(w, E_0, \omega) \left[ p(w, E - \omega) - p(w, E) \right]. \tag{47}$$

This equation only involves convolutions of $p(w, E)$ with a $w$-dependent kernel, and can be solved by Fourier transform.

We define the Fourier transforms

$$k(w,\tau) = \int d\omega \, \sin^2 \frac{\eta(\omega)}{2} e^{-i\omega\tau} = \pi w \left[ L_{-1}(2w\tau) - I_1(2w|\tau|) \right], \tag{48a}$$

$$\tilde{\rho}(w, E_0, \tau) = \int d\omega \, \tilde{\rho}(w, E_0, \omega) e^{-i\omega\tau}, \quad \text{and} \tag{48b}$$

$$p(w, t) = \int dE \, p(w, E) e^{-iEt}, \tag{48c}$$

where we have abused notation by using $\tilde{\rho}$ for both the density of Jacobi decimations and its Fourier transform, and similarly for $p$. In Eq. (48a), the Fourier transform of $\sin^2(\eta/2)$ is evaluated in terms of a modified Struve function $L_n$ and a modified Bessel function $I_n$. The fidelity is $P_0(t) = |p(0, t)|^2$.

Expressing the integral over $\omega$ in terms of Fourier transforms gives

$$-\partial_w p(w, E) = \left( \int \frac{d\tau}{2\pi} e^{iE\tau} [k * \tilde{\rho}](w, E_0, \tau) p(w, \tau) \right) - [k * \tilde{\rho}](w, E_0, 0) p(w, E), \tag{49}$$

where

$$[k * \tilde{\rho}](w, E_0, \tau) = \int \frac{ds}{2\pi} k(w, \tau - s) \tilde{\rho}(w, E_0, s). \tag{50}$$

Fourier transforming Eq. (49) with respect to $E$ reduces the flow equation to an ordinary differential equation:

$$-\partial_w p(w, t) = \left( [k * \tilde{\rho}](w, E_0, t) - [k * \tilde{\rho}](w, E_0, 0) \right) p(w, t). \tag{51}$$

The solution to this equation is an exponential,

$$\log \frac{p(w, t)}{p(w_0, t)} = \int_w^{w_0} dw' \, [k * \tilde{\rho}](w', E_0, t) - [k * \tilde{\rho}](w', E_0, 0) \tag{52a}$$

$$= \int \frac{d\tau}{2\pi} \int_w^{w_0} dw' \, [k(w', t - \tau) - k(w', -\tau)] \tilde{\rho}(w', E_0, \tau). \tag{52b}$$

In the limit of large volume, the density of states diverges, $\nu(E_0) \to \infty$. In the dense regime, where $w_0 = \mathcal{O}\left( \nu(E_0)^{-1/2} \right)$, this implies that the initial size of the decimated element vanishes, $w_0 \to 0$. In this limit, Eq. (52) simplifies further. We have (ignoring resonances as in Sec. 3.2.2)

$$k(w, t - \tau) - k(w, -\tau) = -\pi w^2 (|t - \tau| - |\tau|) + w \, \mathcal{O}\left( (wt)^2, (w\tau)^2 \right), \tag{53}$$

and hence

$$\log \frac{p(0, t)}{p(w_0, t)} = -\frac{1}{2} \int d\tau \, (|t - \tau| - |\tau|) \int_0^{w_0} dw \, w^2 \tilde{\rho}(w, E_0, \tau). \tag{54}$$

Taking the initial state to be an eigenstate of $H_0$, so that $p(w_0, E) = \delta(E - E_0)$, we have $p(w_0, t) = e^{-iE_0 t}$. This phase does not affect the fidelity, $\log P_0(t) = 2\text{Re} \log p(0, t)$. Thus

$$\log P_0(t) = -J^2 \int d\tau \, (|t - \tau| - |\tau|) C_{\text{Jac}}^+(E_0, \tau), \tag{55}$$

where we defined the symmetric *Jacobi autocorrelation function*

$$C_{\text{Jac}}^+(E_0, \tau) = \frac{1}{J^2} \text{Re} \int_0^\infty dw \, w^2 \tilde{\rho}(w, E_0, \tau), \tag{56}$$

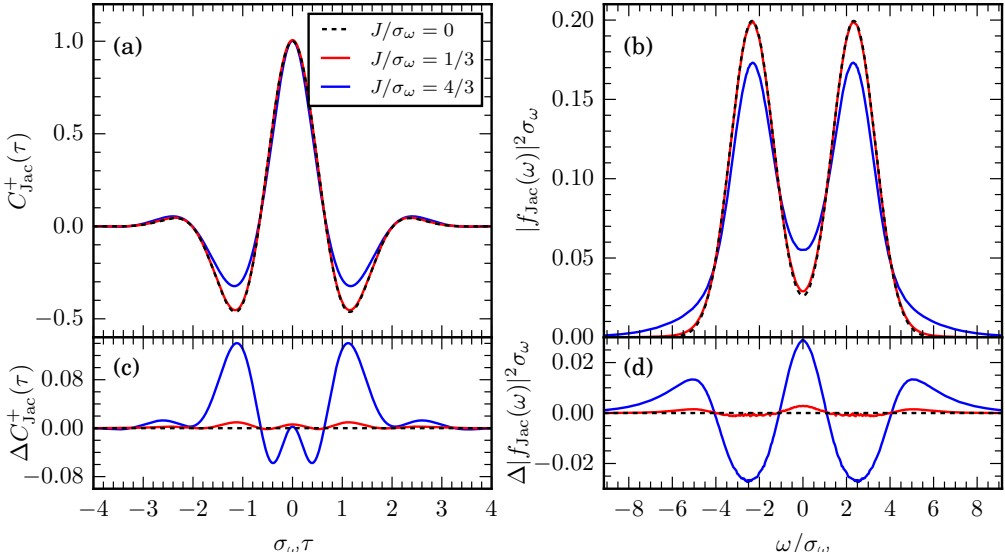

Figure 7: The symmetric (a) Jacobi autocorrelation function $C_{\text{Jac}}^+$ and (b) Jacobi spectral function $|f_{\text{Jac}}|^2$ (blue, red) are corrections to the bare autocorrelation function $C_V^+$ (62) and spectral function $|f_V|^2$ (59) (black dashed) of the perturbation $V$ in $H_0$. For perturbation strengths much less than the characteristic width of the bare spectral function, $J/\sigma_\omega \ll 1$, the difference between the Jacobi autocorrelator and the bare autocorrelator $\Delta C_{\text{Jac}}^+$ (and the difference between the spectral functions $\Delta |f_{\text{Jac}}|^2$) is small (c-d). The difference becomes significant for large perturbations. *Parameters:* As in the random matrix model of Fig. 8.

and have arrived at Eq. (3), our main result.

Note that we replaced the upper limit of the $w$ integral, $w_0$, with infinity. As $w_0$ is the largest decimated element, the density of decimated elements above this cutoff is zero. That is, $\tilde{\rho}(w, E_0, \tau) = 0$ for $w \in (w_0, \infty)$. The integral should be read as a summation over all the Jacobi decimations.

## 4.2 Jacobi spectral function and autocorrelation function

The identification of $C_{\text{Jac}}^+$ with an autocorrelation function is well motivated, as we show below. Indeed, the integral of $w^2 \tilde{\rho}$ is the total norm decimated by the Jacobi algorithm between states of a given energy,

$$\int_0^\infty \mathrm{d}w\, w^2 \rho_{\text{dec}}(w, E, E - \omega) = \int_0^\infty \mathrm{d}w\, w^2 \nu(E) \tilde{\rho}(w, E, \omega) = \nu(E) J^2 |f_{\text{Jac}}(E, \omega)|^2, \quad (57)$$

where we defined a *Jacobi spectral function* $|f_{\text{Jac}}(E, \omega)|^2$. From its definition, the Jacobi spectral function is defined as a sum of squares of matrix elements (up to factors of $\nu(E)$), which makes its interpretation as a spectral function natural. Indeed, if we ignore the rotation of Hamiltonian matrix elements by the Jacobi algorithm, and assume the distribution of decimated elements is the same as the distribution of matrix elements in the original Hamiltonian, we have (Fig. 7)

$$|f_{\text{Jac}}(E, \omega)|^2 = |f_V(E, \omega)|^2 + \mathcal{O}(J^2/\sigma_\omega^3), \quad (58)$$

where

$$|f_V(E_0, \omega)|^2 \mathrm{d}\omega = \sum_{E_j - E_0 \in [\omega, \omega + \mathrm{d}\omega)} |\langle j_0 | V | \psi_0 \rangle|^2, \quad (59)$$

is the usual spectral function for the operator $V$ in $H_0$, with an initial state $|\psi_0\rangle$ of energy $E_0$.

The rotation of the Hamiltonian matrix elements causes the deviation between the Jacobi spectral function and bare spectral function. The error term in Eq. (58) is found by estimating the effect of these rotations at leading order. In one rotation connecting energy differences $\omega$ and $\omega'$, the correction to the spectral function is $\sin^2 \frac{\eta}{2}(|f_V(\omega)|^2 - |f_V(\omega')|^2)$ (as in Eq. (28)), and the total correction is a sum of terms like this from all rotations. In a small angle approximation, $\sin^2 \frac{\eta}{2} \approx w^2/\omega^2$. Replacing $\omega$ by its typical scale $\sigma_\omega$ and performing the sum gives $\sum w^2/\sigma_\omega \approx J^2/\sigma_\omega^2$. Estimating the scale of the initial spectral function by $\sigma_\omega^{-1}$ gives the $\mathcal{O}(J^2/\sigma_\omega^3)$ estimate of the total error.

As the Jacobi algorithm eventually decimates the entire off diagonal, the sum of squares of all decimated elements becomes the norm-squared of the initial perturbation. This implies a sum rule for the Jacobi spectral function,

$$\int dE d\omega \, \nu(E) |f_V(E, \omega)|^2 = \int dE d\omega \, \nu(E) |f_{\mathrm{Jac}}(E, \omega)|^2 = \|V\|_F^2 = \frac{N}{J^2 \beta_0^2}, \tag{60}$$

where $\|\cdot\|_F$ is the Frobenius norm, and we used the definition of the flow time $\beta$ (11).

The symmetric Jacobi autocorrelation function is the (real part of the) Fourier transform of the Jacobi spectral function, so

$$C_{\mathrm{Jac}}^+(E, \tau) = C_V^+(E, \tau) + \mathcal{O}(J^2/\sigma_\omega^2), \tag{61}$$

where

$$C_V^+(E_0, \tau) = \frac{1}{2} \langle \psi_0 | \{V(\tau), V(0)\} | \psi_0 \rangle_c, \tag{62}$$

is the usual connected symmetric autocorrelation function for the perturbation $V$ in the $H_0$ Hamiltonian. (Recall that $V$ was defined to be off diagonal in the $H_0$ eigenbasis, so the appropriate correlation function is the connected one.)

In Appendix A we show that the antisymmetric part of the correlation function is related to a shift in the mean energy of the LDOS. This appears as a phase in the Fourier transform $p(w, t)$, and does not affect the fidelity at leading order in $J/\sigma_E$.

### 4.3 Properties of the solution

#### 4.3.1 Agreement with time-dependent perturbation theory

The log-fidelity can be perturbatively computed using time-dependent perturbation theory (TDPT).[1] Equation (3) correctly reproduces the leading order of TDPT when the Jacobi autocorrelation function, $C_{\mathrm{Jac}}^+(\tau)$, is replaced by the bare autocorrelation function, $C_V^+(\tau)$.

Indeed, substituting Eq. (61) into Eq. (3), we have

$$\log P_0(t) = -J^2 \int d\tau \, (|t - \tau| - |\tau|) \left( C_V^+(E, \tau) + \mathcal{O}(J^2/\sigma_\omega^2) \right) \tag{63a}$$

$$= -4J^2 \int d\omega \, \frac{\sin^2(\omega t/2)}{\omega^2} \left( |f_V(E, \omega)|^2 + \mathcal{O}(J^2/\sigma_\omega^3) \right). \tag{63b}$$

Equation (63b) is the prediction of TDPT for the log-fidelity. Replacing the spectral function by a sum of delta functions,

$$|f_V(E_0, \omega)|^2 = \sum_j |\langle j_0 | V | \psi_0 \rangle|^2 \delta(E_j - E_0 - \omega), \tag{64}$$

---

[1] The same formula for the log-fidelity can also be obtained from the Wigner-Weisskopf approximation [18, Eq. (56)].

produces a potentially more familiar formula in terms of a sum over matrix elements,

$$\log P_0(t) = -4J^2 \sum_j \frac{\sin^2((E_j - E_0)t/2)}{(E_j - E_0)^2} |\langle j_0|V|\psi_0\rangle|^2 + \mathcal{O}(J^4/\sigma_\omega^4). \tag{65}$$

### 4.3.2 Exponential decay at long times

For times $t$ much longer than the decay time of $C_{\text{Jac}}^+$, the solution Eq. (3) predicts exponential decay with $t$.

When $t$ is much larger than the characteristic value of $\tau$ in the integral, we may approximate

$$|t - \tau| - |\tau| \sim |t|, \tag{66}$$

and thus find

$$\log P_0(t) \sim -J^2|t| \int \mathrm{d}\tau \, C_{\text{Jac}}^+(E_0, \tau) = -2\pi J^2|t| |f_{\text{Jac}}(E_0, 0)|^2, \tag{67}$$

provided that $|f_{\text{Jac}}(E_0, 0)|^2$ is finite.

This result is reminiscent of FGR. Indeed, using Eq. (58) we have

$$\log P_0(t) \sim -2\pi J^2|t| \left(|f_V(E_0, 0)|^2 + \mathcal{O}(J^2/\sigma_\omega^3)\right) = -\left(\Gamma_{\text{GR}} + \mathcal{O}(J^4/\sigma_\omega^3)\right)|t|, \tag{68}$$

where $\Gamma_{\text{GR}}$ is the FGR prediction for the decay rate, Eq. (2). The SJA formula recovers FGR at the leading order, and accounts for all corrections in $J/\sigma_\omega$ through the definition of the Jacobi spectral function and autocorrelation function.

If $|f_{\text{Jac}}(E_0, \omega)|^2$ diverges as $\omega \to 0$, then the integral in Eq. (67) does not converge. This scenario should arise when the perturbation $V$ overlaps with a slow hydrodynamic mode, and so decays asymptotically as a nonintegrable power law in time. If the integral in Eq. (67) is regularized by taking its principal value—that is, by using symmetric bounds on the $\tau$ integral—then the result is always finite by Eq. (69) in the next section. However, in this case the decay of $P_0(t)$ can be nonexponential asymptotically.

Note that, for a system with a finite Hilbert space dimension, the exponential decay cannot hold for all times (Sec. 4.3.4, Fig. 2).

### 4.3.3 Quadratic decay at short times

The solution Eq. (3) shows the expected quadratic decrease in fidelity predicted by time-dependent perturbation theory at early times.

To see this, we rewrite the integration kernel in the solution as

$$|t - \tau| - |\tau| = \begin{cases} t, & \text{for } \tau < -|t|, \\ |t| - |\tau| - \text{sgn}(t)\tau, & \text{for } |\tau| < |t|, \\ -t, & \text{for } |t| < \tau. \end{cases} \tag{69}$$

As $C_{\text{Jac}}^+(\tau)$ is even in $\tau$, the integrals for $\tau < -|t|$ and $\tau > |t|$ cancel. Then, the nonvanishing part of the integral is for $|\tau| < |t|$, and for short times we can expand $C_{\text{Jac}}^+(E_0, \tau)$ near $\tau = 0$ as $C_{\text{Jac}}^+(E_0, \tau) = C_{\text{Jac}}^+(E_0, 0) - \mathcal{O}(\tau^2)$. We have

$$\log P_0(t) \sim -J^2 C_{\text{Jac}}^+(E_0, 0) \int_{-|t|}^{|t|} \mathrm{d}\tau \, (|t| - |\tau| - \text{sgn}(t)\tau) = -J^2 C_{\text{Jac}}^+(E_0, 0) t^2, \tag{70}$$

where $J^2 C_{\text{Jac}}^+(E_0, 0) = J^2 \int \mathrm{d}\omega \, |f_{\text{Jac}}(E_0, \omega)|^2$ is thus the energy variance of the LDOS.

One can similarly compute higher order corrections order-by-order from the short time expansion of $C_{\text{Jac}}^+$. For instance, the next order term is $-(J^2 t^4/12)\partial_\tau^2 C_{\text{Jac}}^+(E_0, 0)$.

### 4.3.4 Exponential dependence on volume

In a local many-body system, the fidelity should decrease exponentially in the volume. For example, in a spin system, initial product states are orthogonal to states differing by a single spin flip. Thus, the fidelity should decrease proportionally to the probability of a spin flip occurring on any site. Equation (3) reproduces this behavior.

Indeed, writing the system volume as $v = \mathcal{O}(L^d)$ (where $L$ is the linear size of the system, and $d$ is the dimension), the perturbation $V$ should be extensive,

$$V = \sum_x V_x, \tag{71}$$

where $V_x$ is a (quasi)local operator, and there are $\mathcal{O}(v)$ terms in the sum. In turn, this implies that the connected autocorrelator of $V$ with itself is also extensive,

$$C_V^+(E_0, \tau) = \mathcal{O}(v), \quad \text{and} \quad C_{\text{Jac}}^+(E_0, \tau) = \mathcal{O}(v). \tag{72}$$

Because $\log P_0(t)$ depends linearly on $C_{\text{Jac}}^+(E_0, \tau)$, the fidelity $P_0(t)$ decays exponentially in the volume,

$$\log P_0(t) = \mathcal{O}(v). \tag{73}$$

This implies that there is a cutoff time $t^*$ which is $\mathcal{O}(1)$ in the volume beyond which Eq. (3) no longer holds. Indeed, fidelity decay will be cut off in a system with a finite density of states at a time $t^*$ such that

$$\log P_0(t^*) \approx -S(E_0), \tag{74}$$

where $S(E) \approx \log(J v(E))$ is the entropy at energy $E$ (with $k_B = 1$). We have used the coupling $J$ to fix units in the logarithm for $S(E)$ as this roughly corresponds to the width of the LDOS (Sec. 4.3.3), so that $S(E)$ the the log of the multiplicity, as usual.[2]

In contrast, our analysis assumed a continuum density of states, and so does not recover the saturation of $P_0(t)$. Dividing both sides of Eq. (74) by the volume, we have

$$v^{-1} \log P_0(t^*) = s(E), \tag{75}$$

where $s(E)$ is the (intensive) entropy density. Both sides are $\mathcal{O}(1)$ with the volume in a local many-body system, and so $t^*$ will also be finite in the thermodynamic limit.

The cutoff time may still be very long for weak perturbations, but for sufficiently strong perturbations may even be small enough to preempt the regime of exponential decay. The result in this case is that $\log P_0(t)$ decays quadratically until the cutoff time $t^*$, so that $P_0(t)$ decays as a Gaussian [10].

### 4.3.5 Higher order corrections

Equation (3) can be viewed as the leading order term in an expansion of $\log P_0(t)$ in the small parameter $\epsilon = J/\sigma_E$ at a fixed time $t$. The parameter $\sigma_E$ is the typical energy scale over which the Jacobi autocorrelator $C_{\text{Jac}}^+(E_0, \tau)$ varies in $E_0$. As $E_0$ is extensive, we expect that $\epsilon$ will be very small in a large many-body system.

The higher order corrections in $\epsilon$ are considered in Appendix A. The leading correction is of the form

$$\log P_0(t) = -J^2 \int d\tau \, (|t-\tau|-|\tau|) \, C_{\text{Jac}}^+(\tau) + \mathcal{O}(J/\sigma_E) f(\sigma_\omega t, J/\sigma_\omega), \tag{76}$$

where $f(x, y)$ is quadratic in $x$ for small $x$, and linear in $x$ for large $x$.

---

[2]We have essentially estimated the peak height of the LDOS (the fidelity) with the inverse of its width (related to the diagonal entropy of the initial state). Further, we assumed ETH in replacing the diagonal entropy with the thermodynamic entropy. One could possibly extend this estimate to nonergodic systems by continuing to use the diagonal entropy.

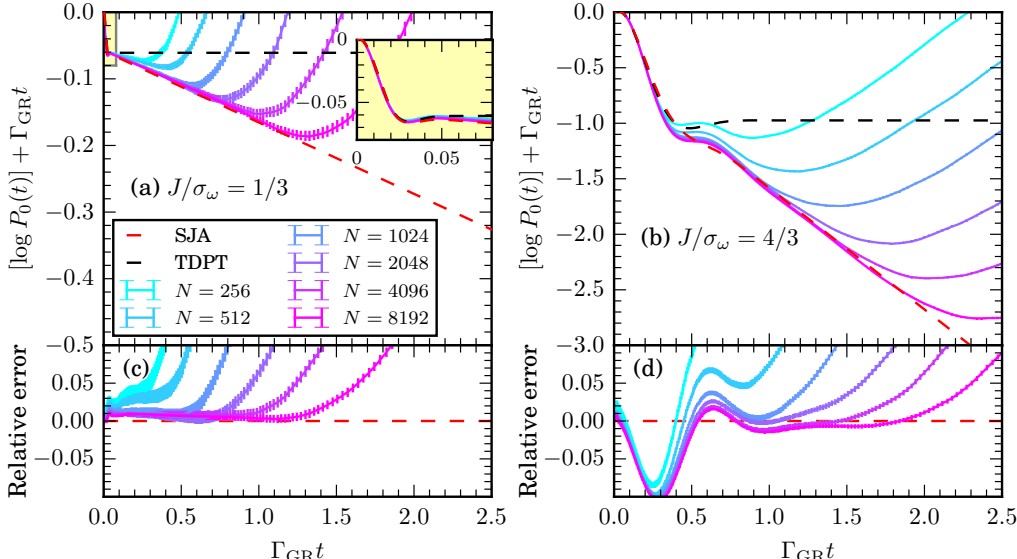

Figure 8: The ensemble averaged log-fidelity, $[\log P_0(t)]$, for the random matrix model (77) is quantitatively reproduced by the SJA formula for large Hilbert space dimension $N$. Even relatively weak perturbations (a) can produce asymptotic decay rates which differ from FGR (black dashed), and stronger perturbations (b) have substantially altered decay rates. These corrections are captured with no free parameters by the SJA formula (red dashed), though there are small errors at early times for large perturbations (c, d). This model exhibits damped oscillations in the log-fidelity (b and inset in a). *Parameters:* $\omega_0/\sigma_E = 0.14$, $\sigma_\omega/\sigma_E = 0.06$, (a, c) $J/\sigma_\omega = 1/3$, (b, d) $J/\sigma_\omega = 4/3$. Logarithms of the fidelity are averaged over $10^4$ random matrix samples, and $C^+_{\text{Jac}}$ used in the SJA formula (3) is computed using $N = 1024$ and $10^3$ samples.

# 5 Numerical verification

Numerical calculations of the fidelity $P_0(t)$ show excellent agreement with the SJA result in Eq. (3) in both random matrix models (Sec. 5.1) and in local many-body Hamiltonians (Sec. 5.2).

In both cases, the numerical procedure is to find an eigenstate of the unperturbed Hamiltonian $H_0$ at a given target energy $E_0$, and then compute $\log P_0(t)$ using exact time evolution under $H = H_0 + JV$. We then compare this to the prediction of the SJA, using the Jacobi algorithm to compute $C^+_{\text{Jac}}(\tau)$. We also compare to the TDPT prediction, using the bare autocorrelator $C^+_V(\tau)$.

The agreement between the prediction Eq. (3) and numerics seems to improve upon averaging the autocorrelation function $C^+_{\text{Jac}}(\tau)$ either over the random matrix ensemble or over eigenstates $|\psi_0\rangle$. Inspecting Eq. (3), the average autocorrelation function should be related to an average log-fidelity, which we denote with square brackets, $[\log P_0(t)]$.

## 5.1 Random matrix model

By explicitly engineering a target spectral function $|f_V(\omega)|^2$ in a random matrix model, we can obtain log-fidelity curves $\log P_0(t)$ which show nontrivial features beyond early time quadratic decay and late time exponential decay. Equation (3) reproduces these features, in addition to

the more generic early and late time behavior.

We consider $N \times N$ random Hamiltonians $H = H_0 + JV$ with matrix elements in a computational basis

$$(H_0)_{jk} = E_j \delta_{jk}, \quad \text{and} \quad V_{jk} = \frac{|f_V(\bar{E}_{jk}, \omega_{jk})|}{\sqrt{\nu(\bar{E}_{jk})}} X_{jk} \quad (j \neq k), \tag{77}$$

where $E_j$ are random numbers drawn independently from a distribution with probability density function $\nu(E)/N$, $|f_V(E, \omega)|^2$ is the target spectral function, and $X_{jk} = X_{kj}$ are drawn from a standard normal distribution. The elements $V_{jk}$ are given by the ETH Ansatz for the matrix elements of a local operator in the eigenbasis of a thermalizing Hamiltonian [4, 50]. The squared Frobenius norm of both $H_0$ and $V$ scale proportionally to the Hilbert space dimensions $N$, while the bandwidth of the model is $\mathcal{O}(1)$.

In this section, and in Fig. 1(a), we take $\nu(E)/N$ to be uniform on $[-\sigma_E/2, \sigma_E/2]$, and

$$|f_V(E, \omega)|^2 = \frac{1}{2\sqrt{2\pi\sigma_\omega^2}} \left[ \exp\left(-\frac{(\omega - \omega_0)^2}{2\sigma_\omega^2}\right) + \exp\left(-\frac{(\omega + \omega_0)^2}{2\sigma_\omega^2}\right) \right], \tag{78}$$

to be an $E$-independent sum of two Gaussians of width $\sigma_\omega$ separated by $2\omega_0$. For the autocorrelation function, this gives

$$C_V^+(E, \tau) = e^{-\sigma_\omega^2 \tau^2/2} \cos(\omega_0 \tau). \tag{79}$$

These functions are shown in Fig. 7 as black dashed curves. Our choice of $\nu(E)$ minimizes the effects of the finite $\sigma_E$ on initial states in the middle of the spectrum. Our choice of $|f_V(E, \omega)|^2$ gives an FGR rate $\Gamma_{\text{GR}} = \sqrt{2\pi}(J^2/\sigma_\omega)e^{-\omega_0^2/2\sigma_\omega^2}$, which can be made small even for large $J^2/\sigma_\omega$ by increasing $\omega_0/\sigma_\omega$. This makes relative deviation from FGR more visible.

A state $|\psi_0\rangle$ with $E = 0$ in $H_0$ is prepared by fixing one of the $H_0$ diagonal elements to zero. To compute the fidelity of this state in a quench to the full Hamiltonian $H = H_0 + JV$, we diagonalize $H$ and compute $P_0(t) = \left|\langle\psi_0|e^{-iHt}|\psi_0\rangle\right|^2$ using a Lehmann expansion. We average $\log P_0(t)$ over the random matrix ensemble. The result is shown in Fig. 1(a) and Fig. 8 for a sequence of Hilbert space dimensions $N$. In these random matrix models, the norm of the perturbation is proportional to $N$, and so $[\log P_0(t)]$ approaches a finite limit as $N \to \infty$ at fixed $t$. This limit curve shows nontrivial behavior, including damped oscillations around the overall decay of the fidelity. This is a consequence of the damped oscillations in the perturbation's correlation function, Eq. (79), which also appear in the logarithm of the fidelity dynamics, Eq. (3).

Indeed, the leading order prediction (which is reproduced by TDPT)

$$\log P_0(t) = -J^2 \int d\tau \, (|t - \tau| - |\tau|)\left(e^{-\sigma_\omega^2 \tau^2/2} \cos(\omega_0 \tau) + \mathcal{O}(J^2/\sigma_\omega^2)\right), \tag{80}$$

may be evaluated in closed form. Defining $\Gamma_{\text{GR}} = 2\pi J^2 |f_V(0,0)|^2 = \sqrt{2\pi}(J^2/\sigma_\omega)e^{-\omega_0^2/2\sigma_\omega^2}$, we have

$$
\begin{aligned}
\log P_0(t) = -\Gamma_{\text{GR}}\Bigg( &\frac{t}{2}\left[\text{erf}\left(\frac{\sigma_\omega t + i\omega_0/\sigma_\omega}{\sqrt{2}}\right) + \text{erf}\left(\frac{\sigma_\omega t - i\omega_0/\sigma_\omega}{\sqrt{2}}\right)\right] \\
&+ 2\frac{e^{\omega_0^2/2\sigma_\omega^2}}{\sqrt{2\pi\sigma_\omega^2}}\left[e^{-\sigma_\omega^2 t^2/2}\cos(\omega_0 t) - 1\right] \\
+ i\frac{\omega_0}{2\sigma_\omega^2}\Bigg[\text{erf}\left(\frac{\sigma_\omega t + i\omega_0/\sigma_\omega}{\sqrt{2}}\right) &- \text{erf}\left(\frac{\sigma_\omega t - i\omega_0/\sigma_\omega}{\sqrt{2}}\right) - 2\,\text{erf}\left(\frac{i\omega_0}{\sqrt{2}\sigma_\omega}\right)\Bigg]\Bigg) + \mathcal{O}(J^4/\sigma_\omega^4),
\end{aligned}
\tag{81}
$$

where erf is the error function, and one can check that the expression is real. This prediction for $\log P_0(t)$ reproduces the generic features expected from Sec. 4.3, and additionally shows the predicted damped oscillations (in the second line).

Even at this order, the nontrivial features of the fidelity are reproduced (Fig. 1 and Fig. 8, black dashed). However, this order of the calculation does not accurately reproduce the rate of asymptotic exponential decay—the numerically observed rate of decay differs from Fermi's golden rule.

To evaluate the higher order corrections, we numerically compute

$$J^2 C_{\text{Jac}}^+(E, \tau) = \frac{1}{\nu(E)} \sum_n w_n^2 (\delta(E - E_{a_n}) + \delta(E - E_{b_n})) \cos((E_{a_n} - E_{b_n})\tau), \tag{82}$$

using the Jacobi algorithm. We bin the energies $E_{a_n}$ and $E_{b_n}$ to compute the dependence on $E$, and average $C_{\text{Jac}}^+(E, \tau)$ over the random matrix ensemble. The integral transform Eq. (3) then provides a prediction for $[\log P_0(t)]$. (Note that using an average $C_{\text{Jac}}^+(E, \tau)$ requires that we average the log of the fidelity.)

We find that this procedure quantitatively reproduces the asymptotic exponential decay, including both the decay rate and the overall scale (which appears in the logarithm as a constant offset). The early time damped oscillations are also reproduced, though with a small error for large perturbation strengths.

Numerically implementing the Jacobi algorithm is more expensive than state of the art exact diagonalization. However, evaluating Eq. (3) using $C_{\text{Jac}}^+(E, \tau)$ computed through the Jacobi algorithm produces curves with smaller error bars for a given number of random matrix samples. Further, the asymptotic exponential decay of $P_0(t)$ is visible with smaller Hilbert space dimensions, as Eq. (3) does not reproduce the asymptotic limit of $P_0(t) \gtrsim 1/N$; it allows for $P_0(t)$ to decay to zero.

## 5.2 Local Hamiltonians

In this section, we compare the SJA formula Eq. (3) to the numerically computed fidelity in several standard local spin chains. We find quantitative agreement, even when the unperturbed spin chain is integrable. This indicates that Eq. (3) is predictive for local Hamiltonians, and not only for random matrix models. While the log-fidelity is also well captured by TDPT in many models, some choices of model show significant deviation from TDPT. These deviations are still captured well by Eq. (3).

We consider two different one-dimensional spin Hamiltonians with periodic boundary conditions: the mixed field Ising model,

$$H_1 = K \left( \sum_{i=1}^{L} \sigma_i^z \sigma_{i+1}^z + g_x \sigma_i^x + h_z \sigma_i^z \right), \tag{83}$$

and XXZ model,

$$H_2 = K \left( \sum_{i=1}^{L} \sigma_i^x \sigma_{i+1}^x + \sigma_i^y \sigma_{i+1}^y + \Delta \sigma_i^z \sigma_{i+1}^z \right), \tag{84}$$

(where $\sigma_i^\alpha$ is a Pauli operator on site $i$) and work in the zero momentum sector of each. While the model (83) is believed to be chaotic for almost all parameter values [56], model (84) is integrable [57–60].

We diagonalize these Hamiltonains for system sizes up to $L = 16$ sites, and average the log-fidelity under the time evolution generated by $H = H_i + JV_{\alpha,\beta}$ for 200 states in the middle

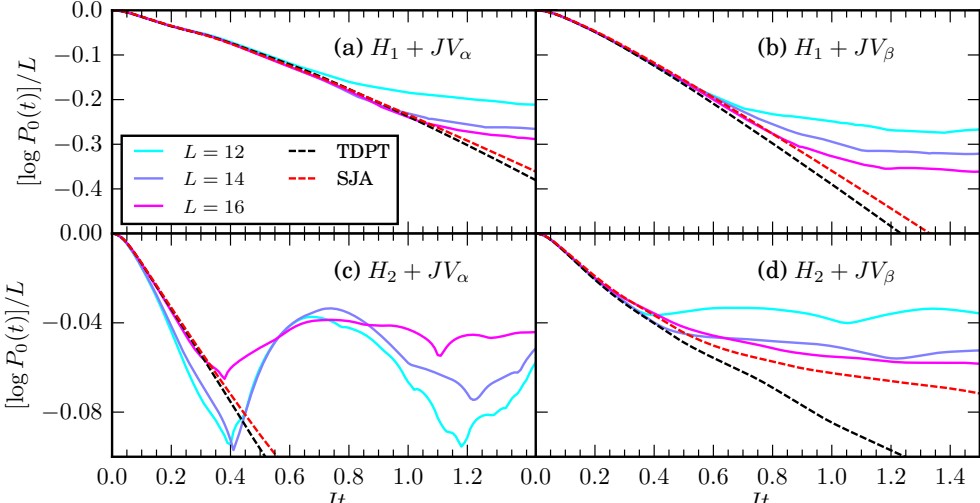

Figure 9: The (intensive) log-fidelity density $[\log P_0(t)]/L$, averaged over 200 states in the middle of the spectrum of a local Hamiltonian (83, 84), is reproduced by the leading order of TDPT, and by the full SJA formula. The SJA formula holds in one-dimensional chaotic models including (a, b) the mixed field Ising model, but also in integrable models such as (c, d) the XXZ model, provided an average over states is made. (c) Both TDPT and SJA only reproduce the fidelity for $L = 16$, which is the system size at which those calculations were performed. The presence of sharp features in the log-fidelity may indicate a dynamical quantum phase transition [54, 55]. *Parameters: $J/K = 0.2$, (a, b) $g_x = 0.9045$, $h_z = 0.8090$, (c, d) $\Delta = 0.5$.*

of the spectrum. We use two different perturbations:

$$V_\alpha = \sum_{i=1}^{L} \sigma_i^x \sigma_{i+1}^x - \sigma_i^y \sigma_{i+1}^y, \tag{85a}$$

$$V_\beta = \sum_{i=1}^{L} \sigma_i^x \sigma_{i+2}^x + \sigma_i^y \sigma_{i+2}^y. \tag{85b}$$

For the nonintegrable Hamiltonian $H_1$, any perturbation should behave similarly, and the choices of $V_\alpha$ and $V_\beta$ are not important. For $H_2$, we have chosen $V_\alpha$ such that it preserves the integrability of the model [61], and $V_\beta$ such that integrability is broken [62].

The results for the fidelity decay are shown in Fig. 9. They show the typical features of the fidelity: they initially decay quadratically, but then become exponential with a decay rate set by FGR. With the extensive perturbations of Eq. (85b), the FGR rate is itself extensive, and it is the log-fidelity density $[\log P_0(t)]/L$ which has a finite thermodynamic limit (Sec. 4.3.4). However, we note that we do not see convergence in $[\log P_0(t)]/L$ for the system sizes in Fig. 9 when the final Hamiltonian is integrable, even before the saturation time $t^*$. This is likely due to more severe finite size effects in integrable and nearly integrable models.

To compare these numerical results to the SJA prediction, we calculate the symmetric, connected autocorrelation function $J^2 C_{\text{Jac}}^+(E_0, \tau)$ of each perturbation in its respective initial Hamiltonian $H_i$ using Eq. (82). We compute the $E_0$ dependence by binning the energies. Equation (3) then gives the prediction for $[\log P_0(t)]$. We additionally compare the exact dynamics to the leading order prediction of TDPT (which is reproduced by the leading order of the Jacobi prediciton). We compute the TDPT expression by explicitly performing the sum over matrix elements given by Eq. (65). For both SJA and TDPT, we use a system size of $L = 16$.

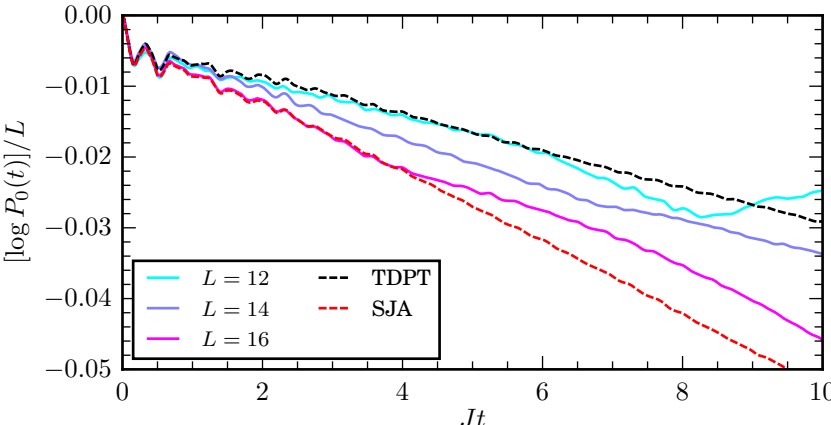

Figure 10: When the perturbing operator $V$ has an oscillatory autocorrelation function in $H_0$, the log-fidelity may fail to be captured by TDPT (black dashed). The SJA formula (3) (red dashed) continues to accurately predict the log-fidelity in this case, even for local Hamiltonians. *Parameters:* $g_x = 0.3$, $h_z = 0.8090$ in the mixed field Ising model $H_1$ (83), with perturbation $V_\gamma = \sum_{i=1}^L \sigma_i^x$, $J/K = 0.1$. The log-fidelity is averaged over 100 states in the middle of the spectrum. As finite size effects are still large at the shown values of $L$ in this model, the TDPT and SJA formula should only be compared to the $L = 16$ system, for which they were calculated.

The initial decay of the fidelity and its subsequent exponential decay tend to be well captured by TDPT for these models and perturbations (up to the cutoff time). The SJA formula performs as well as, or better than, TDPT in each case. However, the discrepancy between the TDPT and SJA formulae is not very large.

This agreement also holds in the model $H_2 + JV_\alpha$ at early times, which is integrable even after the perturbation is introduced. Here, it is important that we are averaging over many states in the middle of the spectrum. The behavior of an arbitrary initial state in an integrable model is not determined just by its energy density, as Eq. (3) would predict. However, sufficient averaging in a small energy window makes $[\log P_0(t)]$ a smooth function of energy [63, 64]. Additionally, the log-fidelity density has a sharp feature, which may be an indication of a dynamical quantum phase transition [54, 55], and beyond which neither TDPT nor the SJA appear to capture the fidelity decay.

Taking lessons from Sec. 5.1, we can engineer models where the discrepancy between TDPT and the SJA is more severe. In Sec. 5.1, the autocorrelation function of the perturbation was oscillatory, which resulted in a strong renormalization of $|f_V(0)|^2$ to $|f_{\text{Jac}}(0)|^2$. In Fig. 10, we quench the transverse field in $H_1$ from a small value $g_x = 0.3$ to a slightly larger value, $g_x + J/K = 0.4$. That is, we consider $H = H_1 + JV_\gamma$ with

$$V_\gamma = \sum_{i=1}^L \sigma_i^x, \tag{86}$$

and $J/K = 0.1$. The initial eigenstates tend to be polarized along the $z$ direction, due the relatively large longitudinal field and ferromagnetic coupling. Then the perturbing $x$ field tends to precess, and produces an oscillatory autocorrelation function. As was the case for the random matrix model, Fig. 10 shows a large discrepancy between the predictions of TDPT at leading order (equivalently, SJA at leading order) and the full SJA formula, Eq. (3). The latter agrees with the numerically exact evolution up to $Jt \approx 4$.

This model is close to an integrable point: $H_1$ with $g_x = 0$. As such, we see large finite size effects, and no convergence of $[\log P_0(t)]/L$ with increasing $L$. The TDPT and SJA calculations

are performed at $L = 16$, and SJA correctly reproduces the fidelity for that system size. TDPT seems to approximate the fidelity for $L = 12$, but this is a coincidence. TDPT performed at $L = 12$ disagrees with the $L = 12$ fidelity.

At all system sizes in Fig. 10, the fidelity seems to deviate from the SJA prediction well before the fidelity saturates. We leave accounting for this feature open to future work.

## 6 Discussion

A statistical description of Jacobi's matrix diagonalization algorithm—the *statistical Jacobi approximation* (SJA)—relates the distribution of decimated elements to observable dynamical properties. Previous work demonstrated how many-body resonances in prethermal many-body localization (MBL) were naturally captured by the SJA, with a corresponding prediction for infinite temperature autocorrelation functions [42]. The finite size scaling of the distribution of decimated elements in prethermal MBL shows that it is part of the *sparse* regime of the SJA description, where the decimated matrix element is much larger than typical matrix elements.

Here, we have provided a statistical description of the opposite *dense* regime, which characterizes the long time dynamics of systems which satisfy the eigenstate thermalization hypothesis (ETH) [4,48–52]. Solving a master equation for the flow of the local density of states (LDOS) under the action of the Jacobi algorithm allows time-dependent fidelities to be calculated after a quench of the Hamiltonian. These results extend beyond the regime of validity of Fermi's golden rule (FGR)—they are applicable for a wider range of time scales, and for larger quench magnitudes. Numerical results show excellent agreement with our predictions.

A natural and useful extension of our results is to predict the behavior of correlation functions in the dense regime using the SJA. Correlation functions of local observables in manybody systems tend to have more structure than the decay of fidelities, making them more difficult to calculate. For example, they reflect the locality of the system's dynamics through light cones [65] and hydrodynamics. As autocorrelation functions continue to decay when the fidelity has saturated, a statistical Jacobi description must maintain control beyond the cutoff time $t^*$ to describe their asymptotic behavior [66].

Calculating the decay of a few-level system coupled to a large environment is independently useful. In addition to being immediately accessible to experiment in several architectures [67–70], this is a prototypical problem for thermalization, with established connections to (prethermal) MBL [38,39,71–74].

As many systems may be described in terms of the decay from an initial state to a continuum of final states, there are several experimental contexts in which our results may be useful. The decay of correlations in Loschmidt echoes can be framed in terms of fidelity [9], with experiments having probed fidelity decay in, for example, nuclear magnetic resonance experiments [75–78], trapped ions [79,80], and classical waves [81]. With new techniques of efficiently measuring fidelity [82], the variety of systems and initial states for which the fidelity can be obtained experimentally is likely to increase.

A particularly interesting context for fidelity is in dynamical quantum phase transitions [54, 55], where $P_0(t)$ develops nonanalyticities in $t$. The application of our formalism to such dynamical transitions and other systems where $P_0(t)$ is not analytic [83,84] is an open direction for future research.

Our methods and results can likely be adapted straightforwardly to periodically-driven (Floquet) systems [85–87]. Floquet systems show dynamical features on several well-separated time scales [5,14,88], which indicate that the SJA could be useful in understanding the behavior of state fidelities in these systems. Further, with their lack of a conserved energy, Floquet

systems represent a minimal setting for understanding the behavior of correlation functions.

Our current analysis involves computing $C_{\text{Jac}}^+(\tau)$ by implementing the Jacobi algorithm. As such, it has the same scaling of numerical effort with system size as exact diagonalization. While we have observed that $C_{\text{Jac}}^+(\tau)$ and the SJA prediction Eq. (3) are more stable with increasing system size than an exact computation of $\log P_0(t)$ at the same size, having an efficient method to compute the Jacobi autocorrelation function would represent a very useful extension to our work.

There appears to be a connection between the SJA and more conventional techniques of random matrix theory. Appendix A indicates a similarity between the Jacobi algorithm's action on diagonal elements of $H$ and the motion of energy levels in Dyson Brownian motion [43, 44, 89]. A precise relationship between the two techniques is currently not clear, but establishing such a connection more concretely may allow some random matrix techniques to be incorporated into the analysis of the Jacobi algorithm. For instance, the joint distribution of eigenvalues and eigenvectors has been calculated for several random matrix ensembles [43, 44]. If the SJA could also be adapted to track such a joint distribution, one could predict state fidelities even beyond the cutoff time $t^*$ at which our current continuum description becomes invalid. Conversely, the application of the SJA to random matrix theory is also an open direction for research.

While both the sparse [42] and dense regimes have been characterized with the SJA, interpolating between these two regimes is an open problem. In prethermal MBL [29–33, 41, 42] or near other dynamical transitions [46] systems may begin in the sparse regime, but then cross over into the dense regime. Understanding this process may reveal important insights into the nature of dynamical transitions in isolated quantum systems.

## Acknowledgments

The authors are particularly grateful to N. O'Dea for pointing out the connection between the leading order of Eq. (3) and TDPT. The authors also thank M. Flynn, A. Kamenev, V. Khemani, C. Laumann, A. Morningstar, A. Polkovnikov, S. Bandyopadhyay, and C. Vanoni for helpful discussions and comments.

**Funding information**   This work was supported by NSF Grant No. DMR-1752759 and AFOSR Grant No. FA9550-20-1-0235 (D.M.L. and A.C.) and by the Laboratory for Physical Sciences (D.M.L.). A.C. and D.M.L. acknowledge the hospitality of MPI-PKS as part of the institute's visitors program. D.H. acknowledges support from the BU CMT visitor fund. This research was supported in part by the National Science Foundation under Grants No. NSF PHY-1748958 and PHY-2309135 (KITP). Numerical work was performed on the BU Shared Computing Cluster, using QuSpin [90, 91].

**Contributions**   D.M.L. derived and solved the flow equation for the LDOS, with important contributions from D.H., M.B., and A.C. D.M.L. performed the random matrix numerics; D.H. and D.M.L. performed the local Hamiltonian numerics. A.C. and M.B. supervised the work on the project. All authors contributed to conceiving the research, interpreting the simulation data, and writing the manuscript.

# A  The flow equation with broad local density of states

In this section we show a more complete derivation of the SJA flow equation for the local density of states (LDOS), including the effects of energy level motion—the changes to the energies $E_{j_n}$ produced by the Jacobi algorithm. The complete flow equation admits an expansion in inverse powers of the energy bandwidth $\sigma_E$, and allows for corrections to the result in Eq. (3) to be computed systematically order-by-order. This extends our results to LDOS which are broader in energy.

## A.1  Energy level motion

In the derivation of the flow equation (23), we neglected the fact that the Jacobi algorithm alters the energy levels $E_{j_n}$. This effect is important when finding corrections to the flow equation.

Upon the decimation of the Hamiltonian matrix element $\langle a_n|H|b_n\rangle$, the energy levels $E_{a_n}$ and $E_{b_n}$ are repelled from each other,

$$E_{a_{n+1}} = \frac{E_{a_n} + E_{b_n}}{2} + \text{sgn}(\omega)\sqrt{\left(\frac{\omega}{2}\right)^2 + w_n^2}, \tag{A.1a}$$

$$E_{b_{n+1}} = \frac{E_{a_n} + E_{b_n}}{2} - \text{sgn}(\omega)\sqrt{\left(\frac{\omega}{2}\right)^2 + w_n^2}. \tag{A.1b}$$

Here, $\omega = E_{a_n} - E_{b_n}$ and $w_n^2 = |\langle a_n|H|b_n\rangle|^2$. The energy update can be expressed as

$$E_{a_{n+1}} - E_{a_n} = \frac{\omega}{2}\left(\sqrt{1 + \left(\frac{2w}{\omega}\right)^2} - 1\right) = \frac{\omega}{2}(\sec\eta(\omega) - 1) = \frac{w^2}{\omega} + O(w^3), \tag{A.2}$$

where the last equality holds in the limit $w \to 0$, to which we specialize, as in the main text. Recall that this is effectively a large volume limit (Sec. 3.2.2).

Equation (A.2) shows that the Jacobi algorithm induces repulsive dynamics between energy levels with kicks that are $1/\omega$ strong. Treating the rotations as stochastic, the Jacobi algorithm thus reproduces Dyson Brownian motion [43, Chapter 4.3]. Rather than stochastically modelling the kicks, we will use the distribution $\rho_{\text{dec}}(w, E, E')$ to encode the statistics of the rotations.

Passing to the continuum from the discrete step Eq. (A.2), as we did for the updates to the wavefunction elements, the sum over energy updates between rotation indices $n$ and $n + \mathrm{d}n$,

$$E_{j_{n+\mathrm{d}n}} - E_{j_n} = \sum_{\substack{m:\, n \le m < n+\mathrm{d}n,\\ a_m = j_m}} \frac{\omega_m}{2}(\sec\eta_m - 1), \tag{A.3}$$

becomes a differential update to $E_{j_n}$. Following an essentially identical procedure to the derivation of Eq. (35) gives the nonlinear ordinary differential equation for $E_{j_n}$. $E(w_n) = E_{j_n}$ is then determined by the decimated element $w_n$ and the initial energy $E(w_0) = E_{j_0}$ through the differential equation

$$-\mathrm{d}_w E(w) = \int \mathrm{d}\omega\, \frac{\omega}{2}(\sec\eta(\omega) - 1)\tilde{\rho}(w, E, \omega) = c(w, E), \tag{A.4}$$

where now

$$\tilde{\rho}(w, E, \omega) = \frac{\rho_{\text{dec}}(w, E, E - \omega)}{\nu(w, E)}, \tag{A.5}$$

involves $\nu(w, E)$, the density of states (DOS) when the decimated element is $w$,

$$\nu(w_n, E) = \sum_j \delta(E - E_{j_n}). \tag{A.6}$$

In writing a differential equation for $E(w)$, we are assuming that each change in energy $E_{j_{n+1}} - E_{j_n}$ is infinitesimally small. From Eq. (A.2), this assumption is valid in the $w \to 0$ limit.

In principle, the differential equation Eq. (A.4) can be solved self-consistently for $E(w)$ by requiring that $\nu(w_0, E_0)$ be related to $\nu(w, E)$ by a change of variables from $E_0$ to $E(w)$. Alternatively, a flow equation for $\nu(w, E)$ can be solved directly. This flow equation is the continuity equation implied by the conservation of energy levels,

$$\mathrm{d}_w \nu(w, E(w)) = \partial_w \nu(w, E(w)) + \partial_E[c(w, E(w))\nu(w, E(w))] = 0, \tag{A.7}$$

which simplifies to

$$-\partial_w \nu(w, E) = \int \mathrm{d}\omega \, \frac{\omega}{2}(\sec \eta - 1)\partial_E \rho_{\mathrm{dec}}(w, E, E - \omega). \tag{A.8}$$

The right hand side is independent of $\nu(w, E)$, and so may be directly integrated to find the solution for the DOS.

## A.2 The flow equation with energy level motion

As the energy levels flow under the action of the Jacobi algorithm, the LDOS flows with them. That is, as the energies $E_{j_n}$ appearing in

$$P_0(t) \approx \left| \sum_j |\langle j_n | \psi_0 \rangle|^2 e^{-iE_{j_n}t} \right|^2 = \left| \int \mathrm{d}E \, p(w_n, E)e^{-iEt} \right|^2, \tag{A.9}$$

are moving, the distribution $p(w_n, E)$ must also move so that the wavefunction weights $|\langle j_n | \psi_0 \rangle|^2$ are being binned in the correct energy interval. As an extreme example, if all the $E_{j_n}$ values move one unit towards positive $E$, then $p(w_n, E)$ must also translate one unit in the same direction.

This feature can be accounted for in the flow equation Eq. (23) by replacing the partial derivative with respect to $w$ by a comoving derivative, similar to that appearing in the continuity equation for the DOS (A.7). The complete flow equation is

$$-\partial_w p(w, E) = \partial_E[c(w, E)p(w, E)] + \int \mathrm{d}\omega \, \sin^2 \frac{\eta(\omega)}{2} \tilde{\rho}(w, E, \omega)$$
$$\times \left[ p(w, E - \omega)\frac{\nu(w, E)}{\nu(w, E - \omega)} - p(w, E) \right], \quad \text{(A.10)}$$

where $\tilde{\rho}$ includes the changing DOS, and is given by Eq. (A.5).

It is convenient to use

$$\tilde{\rho}(w, E, \omega)\frac{\nu(w, E)}{\nu(w, E - \omega)} = \tilde{\rho}(w, E - \omega, -\omega), \tag{A.11}$$

to eliminate the explicit appearance of $\nu(w, E)$ from the flow equation. All such factors are absorbed into $\tilde{\rho}$. The flow equation becomes

$$-\partial_w p(w, E) = \partial_E[c(w, E)p(w, E)] + \int \mathrm{d}\omega \, \sin^2 \frac{\eta(\omega)}{2} \Big[ \tilde{\rho}(w, E - \omega, -\omega)p(w, E - \omega)$$
$$- \tilde{\rho}(w, E, \omega)p(w, E) \Big]. \quad \text{(A.12)}$$

### A.3  Solution to the modified flow equation

We have not found an exact solution to Eq. (A.12), but the equation admits a systematic expansion of the solution in derivatives of the spectral function with respect to $E$.

As before, we Fourier transform this equation to obtain an equation directly for $p(w,t)$, which is the quantity of interest. The transformed equation is

$$-\partial_w p(w,t) = it[c*p](w,t) + \int d\omega \, \sin^2 \frac{\eta(\omega)}{2} \Big[ e^{-i\omega t}[\tilde{\rho}*p](w,t,-\omega) - [\tilde{\rho}*p](w,t,\omega) \Big], \quad \text{(A.13)}$$

where

$$[c*p](w,t) = \int \frac{d\xi}{2\pi} c(w,\xi) p(w,t-\xi), \qquad \text{(A.14a)}$$

$$= \int d\omega \, \frac{\omega}{2}(\sec\eta - 1)[\tilde{\rho}*p](w,t,\omega), \quad \text{and} \qquad \text{(A.14b)}$$

$$[\tilde{\rho}*p](w,t,\omega) = \int \frac{d\xi}{2\pi} \tilde{\rho}(w,\xi,\omega) p(w,t-\xi), \qquad \text{(A.14c)}$$

are convolutions of the Fourier transforms of $c$, $p$ and $\tilde{\rho}$, and we have again used the same symbol for a function and its Fourier transform.

Equation (A.13) can be expressed as a convolution,

$$-\partial_w p(w,t) = \int \frac{d\xi}{2\pi} K(w,\xi,t) p(w,t-\xi), \qquad \text{(A.15)}$$

where the kernel is

$$K(w,\xi,t) = \int d\omega \left[ \sin^2 \frac{\eta}{2}(e^{i\omega t}-1) + \frac{i\omega t}{2}(\sec\eta - 1) \right] \tilde{\rho}(w,\xi,\omega) \qquad \text{(A.16a)}$$

$$\sim \int d\omega \left[ \frac{1}{\omega^2}(e^{i\omega t}-1) + \frac{it}{\omega} \right] w^2 \tilde{\rho}(w,\xi,\omega), \qquad \text{(A.16b)}$$

where the second equality holds for $w \to 0$. Alternatively, expressing the $\omega$ integral in terms of the Fourier transformed functions,

$$K(w,\xi,t) = -\frac{1}{2} \int d\tau \, (|t+\tau| - |\tau| + t \, \mathrm{sgn}(\tau)) w^2 \tilde{\rho}(w,\xi,\tau). \qquad \text{(A.17)}$$

The width of the kernel $K(w,\xi,t)$ in $\xi$ is assumed to be narrow. In a many-body Hamiltonian, it relates to the variation in the spectral function in the energy $E$. As energy is an extensive variable, this variation should be suppressed in the volume. Thus, we can expand $p(w,t-\xi)$ in the small parameter $\xi$. In fact, $l(w,t-\xi) = \log p(w,t-\xi)$ will be better behaved.

In terms of a typical energy scale $\sigma_E$, we introduce the $\mathcal{O}(1)$ dimensionless time $\zeta = \sigma_E \xi$. Further, we introduce the rescaled kernel

$$\epsilon K'(w/J,\zeta,t) = K(w,\zeta/\sigma_E,t), \qquad \text{(A.18)}$$

where $\epsilon = J/\sigma_E$ is a small parameter.

The flow equation for $l(w,t)$ is then

$$-\partial_{w/J} l(w,t) = \epsilon \int \frac{d\zeta}{2\pi} K'(w/J,\zeta,t) \exp[l(w,t-\zeta/\sigma_E) - l(w,t)]. \qquad \text{(A.19)}$$

The solution to this equation can be calculated order-by order in $\epsilon$, as the right hand side is smaller by a factor $\epsilon$ than the left. In more detail, define a sequence $l_k(w, t)$ for $k \geq -1$ by setting the first member of the sequence to the initial condition for $l(w, t)$,

$$l_{-1}(w, t) = l(w_0, t) = -iE_0 t, \tag{A.20}$$

and iteratively define subsequent $l_k(w, t)$ as the solution to the initial value problem

$$-\partial_{w/J} l_{k+1}(w, t) = \epsilon \int \frac{\mathrm{d}\zeta}{2\pi} K'(w, \zeta, t) e^{l_k(w, t - \zeta/\sigma_E) - l_k(w, t)},$$

$$\text{with initial conditions} \quad l_{k+1}(w_0, t) = l(w_0, t). \tag{A.21}$$

We show below that this implies $l_{k+1}(w, t) - l_k(w, t) = \mathcal{O}(\epsilon^{k+1})$. For small $\epsilon$ and $k \to \infty$, we have $l_{k+1}(w, t) \approx l_k(w, t)$, which implies that $l_{k+1}(w, t)$ approximately satisfies Eq. (A.19). That is, the sequence $l_k(w, t)$ gives an asymptotic sequence approaching the actual solution.

The $l_0(w, t)$ member of the sequence is the solution found in the main text. Substituting Eq. (A.20) into Eq. (A.21) to find $l_0(w, t)$ produces the result from the main text, Eq. (54), but with additional terms which accounts for the drift of energy levels.

$$-\partial_w l_0(w, t) = \sigma_E^{-1} \int \frac{\mathrm{d}\zeta}{2\pi} K'(w, \zeta, t) e^{i\zeta E_0/\sigma_E} = K(w, E_0, t). \tag{A.22}$$

Writing out the kernel completely, and integrating with respect to $w$,

$$l_0(w, t) = -iE_0 t - \frac{1}{2} \int \mathrm{d}\tau \, (|t + \tau| - |\tau| + t \operatorname{sgn}(\tau)) \int_w^{w_0} \mathrm{d}w' \, w'^2 \tilde{\rho}(w', E_0, \tau). \tag{A.23}$$

The log-fidelity is $\log P_0(t) = l(0, t) + l(0, t)^*$, where $l(0, t)^*$ is the complex conjugate of $l(0, t)$. For $l_0(w, t)$, this gives

$$\log P_0(t) = -J^2 \int \mathrm{d}\tau \, (|t + \tau| - |\tau| + t \operatorname{sgn}(\tau)) \operatorname{Re}[C_{\text{Jac}}(E_0, \tau)], \tag{A.24}$$

where

$$C_{\text{Jac}}(E_0, \tau) = J^{-2} \int_0^\infty \mathrm{d}w \, w^2 \tilde{\rho}(w, E_0, \tau) = C_{\text{Jac}}^+(E_0, \tau) + iC_{\text{Jac}}^-(E_0, \tau), \tag{A.25}$$

and $\tilde{\rho}(w, E_0, \omega)$ being real valued implies

$$C_{\text{Jac}}^+(E_0, \tau) = C_{\text{Jac}}^+(E_0, -\tau), \quad \text{while} \quad C_{\text{Jac}}^-(E_0, \tau) = -C_{\text{Jac}}^-(E_0, -\tau). \tag{A.26}$$

The antisymmetric part of $C_{\text{Jac}}$ is imaginary, and so does not appear in $\log P_0(t)$ at this order. As such, the antisymmetric $\operatorname{sgn}(\tau)$ part of the kernel can be neglected. (It causes a shift in the mean energy of the LDOS, which does not affect the fidelity.) As the remaining $C_{\text{Jac}}^+(E_0, \tau)$ term is symmetric, we can replace $\tau \to -\tau$ in the transformation and obtain

$$\log P_0(t) = -J^2 \int \mathrm{d}\tau \, (|t - \tau| - |\tau|) C_{\text{Jac}}^+(E_0, \tau), \tag{A.27}$$

which is Eq. (3).

We complete this section by proving $l_{k+1}(w, t) - l_k(w, t) = \mathcal{O}(\epsilon^{k+1})$ using induction. The expressions Eqs. (A.22) and (A.23) demonstrate that $l_0(w, t) - l_{-1}(w, t)$ is $\mathcal{O}(1)$ with $\epsilon$, providing the induction base. Assuming

$$\Delta l_k(w, t) = l_k(w, t) - l_{k-1}(w, t) = \mathcal{O}(\epsilon^k), \tag{A.28}$$

we have

$$-\partial_{w/J}\Delta l_{k+1}(w,t) = \epsilon \int \frac{\mathrm{d}\zeta}{2\pi} K'(w,\zeta,t) \left( e^{l_k(w,t-\zeta/\sigma_E)-l_k(w,t)} - e^{l_{k-1}(w,t-\zeta/\sigma_E)-l_{k-1}(w,t)} \right) \quad \text{(A.29a)}$$

$$= \epsilon \int \frac{\mathrm{d}\zeta}{2\pi} K'(w,\zeta,t) e^{l_{k-1}(w,t-\zeta/\sigma_E)-l_{k-1}(w,t)} \left( e^{\Delta l_k(w,t-\zeta/\sigma_E)-\Delta l_k(w,t)} - 1 \right)$$

$$\text{(A.29b)}$$

$$= \mathcal{O}\!\left(\epsilon^{k+1}\right), \quad \text{(A.29c)}$$

where we used Eq. (A.28) in the last line to replace $e^{\Delta l_k} - 1 = \mathcal{O}\!\left(\epsilon^k\right)$. Integrating with respect to $w/J$ and using the initial condition $\Delta l_{k+1}(w_0, t) = 0$ gives the desired result.

## A.4 Leading correction

The iterative method of Sec. A.3 can be used to evaluate the error term in Eq. (3) in terms of $\rho_{\mathrm{dec}}(w, E_1, E_2)$. This allows the behavior of the error term at short and long times to be deduced. In this section, we explicitly perform this calculation, and obtain the results summarized in Sec. 4.3.5.

First, we observe that the energy level motion caused by the Jacobi algorithm produces an $\mathcal{O}(\epsilon)$ correction to the density of states: $\nu(w, E) = \nu_0(E) + \epsilon \nu_1(w, E)$, where $\nu_0$ is the DOS of $H_0$. Integrating Eq. (A.8), we find

$$\nu(w,E) = \nu_0(E) + \epsilon \int_w^{w_0} \mathrm{d}w' \int \mathrm{d}\omega \, \frac{\omega}{2J} (\sec\eta - 1) \partial_{E/\sigma_E} \rho_{\mathrm{dec}}(w', E, E - \omega). \quad \text{(A.30)}$$

We introduced typical scaling factors $\sigma_\omega$ for $\omega$ and $\sigma_E$ for $E$ to emphasize that the correction is smaller by a factor of $\epsilon$ than the leading term (assuming $J$ and $\sigma_\omega$ are similar). Note that the integral of the correction over $E$ is zero, as it is the total derivative of a function which decays to zero at large and small $E$.

The term which appears in the solution for $l(w, t)$ is $\tilde\rho(w, E, \omega)$, which we can expand in terms of $\tilde\rho_0(w, E, \omega) = \rho_{\mathrm{dec}}(w, E, E - \omega)/\nu_0(w, E)$ as

$$\tilde\rho(w,E,\omega) = \frac{\rho_{\mathrm{dec}}(w,E,E-\omega)}{\nu(w,E)} \quad \text{(A.31a)}$$

$$\approx \tilde\rho_0(w,E,\omega)\left(1 - \epsilon \frac{\nu_1(w,E)}{\nu_0(E)}\right) \quad \text{(A.31b)}$$

$$= \tilde\rho_0(w,E,\omega)\left(1 - \epsilon \int_w^{w_0} \mathrm{d}w' \int \mathrm{d}\omega' \, \frac{\omega'}{2J} (\sec\eta - 1) \partial_{E/\sigma_E} \tilde\rho_0(w',E,\omega')\right). \quad \text{(A.31c)}$$

Observe that the higher order corrections to $\tilde\rho(w, E, \omega)$ are independent of $\omega$. This makes including the correction to $l_0(w, t)$ straightforward. Expanding Eq. (A.23), we have

$$l_0(w,t) = -iE_0 t - \frac{1}{2} \int \mathrm{d}\tau \, (|t+\tau| - |\tau| + t\,\mathrm{sgn}(\tau)) \int_w^{w_0} \mathrm{d}w' \, w'^2 \tilde\rho_0(w',E_0,\tau)$$

$$\times \left(1 - \epsilon \int_{w'}^{w_0} \mathrm{d}w'' \int \mathrm{d}\omega' \, \frac{\omega'}{2J} (\sec\eta - 1) \partial_{E/\sigma_E} \tilde\rho_0(w'',E_0,\omega')\right). \quad \text{(A.32)}$$

The additional integral terms are complicated, but independent of $\tau$. Thus, all the properties from Sec. 4.3 continue to hold for the corrected solution $l_0(0, t) + l_0(0, t)^*$. The early time expansion will be quadratic, and the late time behavior will be linear, though the coefficients in these expansions will be slightly modified.

We also need to account for the $\mathcal{O}(\epsilon)$ correction in $\Delta l_1(w,t)$. Working to $\mathcal{O}(\epsilon)$ in Eq. (A.29b), we have

$$-\partial_{w/J}\Delta l_1(w,t) \approx \epsilon \int \frac{\mathrm{d}\zeta}{2\pi} K'(w,\zeta,t) e^{i\zeta E_0/\sigma_E} \left[\Delta l_0(w,t-\zeta/\sigma_E) - \Delta l_0(w,t)\right]. \tag{A.33}$$

Making a short time expansion for the term in square brackets and using Eq. (A.22), we have

$$-\partial_{w/J}\Delta l_1(w,t) \approx \epsilon \int \frac{\mathrm{d}\zeta}{2\pi} K'(w,\zeta,t) e^{i\zeta E_0/\sigma_E}(-\sigma_E^{-1}) \int_w^{w_0} \mathrm{d}w'\, \partial_t K(w',E_0,t) \tag{A.34a}$$

$$= -i\epsilon \partial_{E_0/\sigma_E} K(w,E_0,t) \int_w^{w_0} \mathrm{d}w'\, \partial_t K(w',E_0,t), \tag{A.34b}$$

$$\Delta l_1(0,t) = -i\epsilon \int_0^{w_0} \mathrm{d}w\, \partial_{E_0/\sigma_E} K(w,E_0,t) \int_w^{w_0} \mathrm{d}w'\, \partial_{Jt} K(w',E_0,t) + \mathcal{O}(\epsilon^2). \tag{A.34c}$$

The last line is written in terms of dimensionless combinations (the integral of $K$ with respect to $w$ is dimensionless).

One may expand the kernels in Eq. (A.34c) to obtain an expression in terms of $\tilde{\rho}_0$, but the expression as written is already sufficient to deduce the short and long time behavior of $\Delta l_1(0,t)$. The kernel $K(w,E_0,t)$ has a quadratic real part and linear imaginary part at short times. At long times, both its real and imaginary parts are linear in $t$. It is then straightforward to check from Eq. (A.34c) that $\Delta l_1(0,t)$ shares these properties. Its real part is quadratic at short times and linear at long times, while its imaginary part is linear at both short and long times. In principle, the coefficients in these expansions can be calculated from Eq. (A.34c) in terms of an integral involving $\tilde{\rho}_0$.

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
