# Peer review of "Beyond Fermi's golden rule with the statistical Jacobi approximation"

_SciPost Physics, doi:SciPost Phys. 15, 251 (2023)_

## Round 3 · Referee Report · Anonymous (Referee 1) · 2023-10-7

Report

This paper gives a thorough treatment of the authors' so-called statistical Jacobi approximation (SJA). This is a method for approximating the fidelity decay of a quantum state in a high-dimensional Hilbert space under certain classes of quantum dynamics that are sufficiently quantum chaotic. The method has been used in Ref 39 to obtain some nice new insights about the crossover to the prethermal MBL regime in some MBL (many-body localization) models. Thus this more detailed exposition of this method may be useful if others want to use this method for other systems or questions.

I have not had the patience or stamina to go through this in much detail. But to the extent I have tried to absorb some of it, I have the following suggestions:

In Eq. (3), the "error" is suggested to be of order (J/sigma_E). But certainly this error must vanish in the limit of time t=0. Can the error be stated in a more informative way that indicates how it behaves vs time?

In Eq. (64), the stated "error" is dimensionally incorrect, so this needs to be fixed. Here again, this error must vanish for t=0, so again can it be stated in a way that also gets this correct?

In Eq. (69) you put \sim, as if there is an unknown prefactor that you can not calculate. But this is "just" lowest-order perturbation theory, so I would think you could get the order t^2 term here exactly using TDPT (?). Since you have stated the "error" in much of the rest of the paper, here you should also show it. Is it just the next order terms in perturbation theory (at t^3 if we do not have time-reversal symmetry and maybe at t^4 if we do)?
  • validity: good
  • significance: good
  • originality: high
  • clarity: good
  • formatting: -
  • grammar: -

Author:  David Long  on 2023-11-21  [id 4137]

(in reply to Report 1 on 2023-10-07)

We thank the Referee for their report and suggested changes. We believe that all of their comments have been addressed in the revised manuscript, and that the Referee should now find the paper appropriate for publication.

We respond to the Referee's specific comments below.

1: The error term in Eq. (3). We use $O(J/\sigma_E)$ in Eq. (3) to denote the scaling of the error at fixed $t$. A more complete statement of the error would be $[\log P_0(t)] = ... + O(J/\sigma_E) f(\sigma_\omega t)$, where$ f(x)$ is finite for $J/\sigma_E \to 0$, behaves as $f(x) \sim f''(0) x^2/2$ for short times, and grows linearly in $x$ for large $x$. This leading correction can in principle be calculated using the framework in Appendix A. We have included enough of this calculation in the new appendix A.4 to verify the statements we have made here, and summarize this information in the new Sec. 4.3.5.

2: Dimensions in Eq. (64). The error term has been corrected. It is meant to be $O(J^4/\sigma_\omega^4)$, as one should expect from the next order of TDPT. The time dependent correction here is again of the form $(J^4/\sigma_\omega^4) g(\sigma_\omega t)$, where $g(x)$ is a dimensionless function which is quadratic for small $x$, linear for large $x$, and finite for $J/\sigma_\omega \to 0$.

3: "$\sim$" symbol in Eq. (69). We are using "$\sim$" in the sense defined by, for instance, this article. That is, Eq. (69) is correct for $t \to 0$, even up to constant factors. The error would be given by the next order in a short time expansion of $C_{\mathrm{Jac}}^+(\tau)$. This is $-(J^2 t^4/12) \partial_\tau^2 C_{\mathrm{Jac}}^+(E_0, 0)$, and we have included this comment in the revised manuscript. We also corrected an error in this equation. The $\mathrm{sgn}(t)|\tau|$ term should have been $\mathrm{sgn}(t)\tau$, with no absolute values, as in the previous equation.

---

## Round 3 · Referee Report · Ivan Khaymovich (Referee 2) · 2023-10-24

Strengths

1 - Analytical method to calculate the fidelity (return or survival probability)
2 - Quantitative agreement with the numerical data in considered random-matrix and many-body systems.

Weaknesses

1 - Computation complexity of Jacobi algorithm,
2 - Necessity of exact diagonalization in order to calculate the autocorrelation function (like in (79)).

Report

In the manuscript "Beyond Fermi’s golden rule with the statistical Jacobi approximation", the authors consider an approximate analytical calculation of the fidelity (return or survival probability) in a generic random-matrix or many-body system under the perturbation and relate it to the Jacobi rotation algorithm and the corresponding autocorrelation function.

The resulting formula (3) matches well the data for random-matrix as well as integrable and chaotic many-body examples, even if the Fermi's Golden rule result fails.

The manuscript is clearly written and brings an interesting idea of the relation between the fidelity and a certain autocorrelation function to the audience. At the same time, I have a bunch of questions and comments to the authors about the manuscript:

1) How far can one extend the suggested method to the systems with anomalous relaxation (beyond exponential decay of the fidelity)? As examples of such systems, one can name a few: - Power-law random banded matrix model (PLRBM), - Random-regular graph (RRG), - Log-normal or Levy Rosenzweig-Porter (LNRP, Levy RP) models. In all such models, the return probability at long times decays either polynomially (PLRBM) or stretch-exponentially (RRG, LNRP, Levy RP) in a range of parameters. Please comment on the possible extensions in the main text, not only as a side remark in page 19. In this sense, the references [12, 13] in page 4 may be supplemented by other works on dynamics in random-matrix models and random graphs: see, e.g., https://doi.org/10.1088/1751-8121/aa77e12 https://doi.org/10.21468/SciPostPhys.11.2.045 and many others.

2) In addition to the previous question, it is of high interest to describe the dynamics in disordered systems with the integrability, emergent (like in many-body localization) or built-in (like in Bethe-ansatz integrable models). Can you say something about the fidelity decay in such cases more generally, but not only on the XXZ Heisenberg chain example? Can the method predict the deviations from the exponential decay or revivals? If not, what are the main limitations of the method?

3) Please compare the suggested method with the works by Lea F. Santos ([3, 10, 11, 56] and several others), where she with coauthors analyzed in many detail the fidelity decay. What are the main differences with your method? Can one see a correlation hole and other attributes of ergodic systems with the suggested Jacobi method?

4) The authors claim and demonstrate that their method goes beyond Fermi's Golden rule result. What about the comparison with more involved Wigner-Weisskopf approximation (see, e.g., a work by Monthus http://dx.doi.org/10.1088/1751-8121/aa77e1 and many others on Levy and LN RP)?

5) The validity conditions of the method, though being summarized somehow in the Table 1, are still rather spread and unclear to me. Is it possible to summarize them partially graphically and clarify the limitations of the method?

6) In Eq. (73) the authors assume ergodicity of the system by placing the saturation point of the fidelity to be an exponential of the entropy. Is it possible to extend this to the non-ergodic phases of matter within the same methodology? Please add the corresponding discussion in the main text.

7) In the conclusion, the authors claim that the Jacobi method has much in common with Dyson Brownian motion. What about generalized Dyson Brownian motion methods, like in https://doi.org/10.21468/SciPostPhys.11.6.101? Is it possible to implement them to the analogous algorithms (maybe related to Shermann-Morisson formula instead of Jacobi rotations)? Please clarify this in the main text.

8) The phrase "Reference [25] defined a similar method, but did not identify a relation to the Jacobi algorithm" is formulated a bit weirdly. Indeed, the method in [25] has the same origin as in [20-22], based on the standard method of a strong-disorder renormalization group. If the authors would like to comment on [25] by comparing the methods, I suggest them to discuss the relation the above renormalization group method to the Jacobi method in general, first.

Requested changes

0 - Please address all the questions of the above report by adding the corresponding discussions to the main text.

1 - Fig. 1 (b): it seems that at long times the L=16 data shows the exponential decay, which is closer to the TDPT, but not to the SJA. Please clarify this issue.

2 - Fig. 3 (b, c): I am not sure that I understand how the distribution function can be parametrically large with the Hilbert space dimension. Please clarify the definition and normalization of $\rho_dec$.

3 - Eq. (13): It seems that the factor of 2 in the exponent is missing between the parts, separated by $\Rightarrow$. Please clarify this issue and the correspondence to Fig. 4.

4 - $\eta(\omega)$ seems to be undefined in Eq. (23). Please define it before using.

5 - When and for which systems the assumption "Each of these steps requires that $E_{jn}$ and $|\langle j_n|\psi_0\rangle|^2$ be slowly varying with n to be valid" is valid?

6 - Discussion before Eq. (33): "We assume that the cross term in Eq. (28) averages to zero." In order to be clear, one should compare the higher-order averages of cross terms with the diagonal ones. Please provide this information in the text.

7 - Eq. (52) and other places: Please write explicitly the conditions for (52) and other assumptions: for (52) should it be $w t \ll 1$?

8 - Fig. 7 (bottom row): Better to show the relative error as the absolute error does not provide a valuable information.

9 - "Replacing the sum over all $sin^2 \frac{\eta}2$ by $J^2/\sigma^2_{\omega}$" - please clarify when and how one can do it.

10 - Sec. 5, second paragraph: Why should I consider $C_{Jac}$ and $C_V$ in different cases? How should one choose?

11 - Eq. (75) : the suggested random-matrix model is unknown. Please provide some properties of the model and links to the known examples.

12 - Fig. 8: the system sizes, used for the numerical simulations are rather small ($2^{13}$ with respect to $2^{16}$ in the many-body examples). Please clarify why it is not needed to increase the system sizes or increase it in the simulations up to the state-of-the-art for random-matrix models.

13 - Eq. (76) and discussion after it: the peaks are separated by $2\omega_0$, but not by $\omega_0$. Factor of $2$ is missing.

14 - the damped oscillations, claimed in page 22, are nearly invisible. Please consider to change parameters to make it visible.

15 - What is the physical origin of the discrepancy between Fermi's Golden rule results and Jacobi method, discussed in page 22? Please clarify in the text.

16 - Eq. (79): it seems that the Jacobi method needs the information about the eigenvalues for the calculation of the autocorrelation function. Please clarify the complexity of the Jacobi method calculation with respect to the direct ED-calculations.

17 - The footnote in page 22 is lengthy and complicated. Please consider to place it to the main text or to the Appendix. Please make it clear in the text.

18 - Fig. 9 (related to the item 16): the authors claim the absence of fitting parameters in the Jacobi method, but in the caption to Fig. 9 and in Eq. (79) the autocorrelation function is calculated for a certain fixed system size. Does it mean that at least a system size $L$ is a kind of the "fitting" parameter, which is needed for the Jacobi-method calculations?

19 - Fig. 9 shows nearly no difference between SJA and TDPT. What is the reason of showing this plot and to use SJA there? Are SJA calculations faster?

  • validity: high
  • significance: good
  • originality: top
  • clarity: good
  • formatting: excellent
  • grammar: perfect

Author:  David Long  on 2023-11-21  [id 4136]

(in reply to Report 2 by Ivan Khaymovich on 2023-10-24)

We thank Prof. Khaymovich for his detailed report. We have incorporated most of his comments and suggested changes into the revised manuscript, and believe that they help improve the paper. Those suggested changes which we have not included, we will directly address below.

We address Prof. Khaymovich's specific comments in the order he made them. We will number our responses as he did, beginning with 0 (with 8 sub-points) and ending with 19.

0.1: Systems with anomalous relaxation, other than exponential decay. All the systems we know of with this property (including those listed by Prof. Khaymovich) are in the sparse regime. The Frobenius norm of a row or column in the Hamiltonian for these models is dominated by a vanishing fraction of the elements. Our current manuscript only concerns the dense regime, but we expect that the analysis of our previous paper, Ref. [39] (now Ref. [42] in the revised version), should apply in some parameter regime of those models. (Although, in the context of (generalized) Rosenzweig-Porter models, the ratio of the Frobenius norm of the off diagonal to that of the diagonal can also vanish. We have not considered this scenario in either this paper or Ref. [39].) The remark on page 19 concerned a different scenario, where the model remains in the dense regime, but the spectral function of the perturbation diverges at small frequencies. We have not analyzed this case, nor the case where the spectral function vanishes at small frequencies, so we do not comment on these scenarios in detail. We have included a comment summarizing the above discussion to section 2.2, and included the recommended references.

0.2: Applicability to other integrable models. Our analysis should apply to any model which begins in the dense regime of the SJA. While exceptions exist, our understanding is that most integrability-breaking perturbations of integrable models tend to be in the dense regime. (One important exception is nearly localized systems, for which local perturbations are sparse.) Our analysis will always predict asymptotic decay of the fidelity, though there can be more complicated dynamics at short times. Revivals should require some phase coherence in the evolution, which we expect to be washed out at large system sizes in the dense regime. There is likely a broad array of behavior which can occur for integrability preserving perturbations. In this case, our analysis could only apply to an initial microcanonical state in the quench, not actual eigenstates.

0.3: Comparison with Lea Santos. Much of the analysis of Santos et al. is focused on long times, beyond the cutoff time $t^*$. For instance, the correlation hole is a feature of the turnover from the intermediate time decay of the fidelity to the late time saturation. Our analysis does not capture any feature at or beyond $t^*$, but we believe it should describe the system well up to that cutoff. The discussion of generic features of the fidelity in the introduction should correctly represent the conclusions of Prof. Santos's work, and already includes citations to her papers on the topic. In the revised manuscript, we more explicitly point out that the "dip and ramp" feature near $t^*$ corresponds to the correlation hole.

0.4: Comparison with the Wigner-Weisskopf approximation. As we understand it, the Wigner-Weisskopf approximation refers to a particular analysis of fidelity decay where a kind of Markovian approximation is made, which allows simple differential equations for the log-fidelity to be derived. The analysis we have referred to as "TDPT" reproduces the same result as the Wigner-Weisskopf approximation as the leading order of perturbation theory for the log-fidelity. Explicitly, the real part of Eq. (56) in the Monthus reference cited by Prof. Khaymovich should be compared to Sec. 4.3.1 in our manuscript. Thus, the comparison of the SJA to the Wigner-Weisskopf approximation is the same as its comparison to TDPT. Qualitative features are reproduced by TDPT/Wigner-Weisskopf, but the SJA shows quantitative improvement for larger perturbations. In the Monthus reference, careful attention is paid to the competition between the broadening of the LDOS compared to the level spacing. No such competition exists in the models we consider, where the level spacing is asymptotically much smaller than the LDOS width, justifying the continuum approximation for the LDOS. As such, the unusual features of fidelity decay that Monthus uncovers with the Wigner-Weisskopf approximation do not appear in our analysis.

0.5: Validity conditions for the assumptions in Table 1. We have changed Table 1 to include a column stating the interpretation of each approximation. We hope that this provides more context for when each approximation can be expected to hold. As stated in the caption, all approximations should be valid for a large many-body system which satisfied the eigenstate thermalization hypothesis, and at times before the cutoff time $t^*$.

0.6: Saturation point for the fidelity. One might attempt to place the saturation point of the fidelity by comparing the log-fidelity to the diagonal entropy of the initial state in the Hamiltonian eigenbasis. This is essentially what we have done, with an additional assumption of ergodicity to pass from the diagonal entropy to the usual thermodynamic entropy at the specified energy density. Even in the ergodic case, this prediction for the saturation time is not precise. Roughly, we are estimating the peak height of the LDOS (saturated fidelity) from its width (entropy). Still, this should give a good heuristic. We have added this discussion to a footnote on page 20 of the revised manuscript.

0.7: Comparison to Dyson Brownian motion. The comparison of Jacobi to Dyson Brownian motion is currently unexplored. One may observe that the Jacobi algorithm causes levels connected by a matrix element $w$ to repel with a kick which is, for $w\ll \omega$, given by $w^2/\omega$, as is the case for Dyson Brownian motion. However, this is the extent to which we understand the two processes are related. As such, we are not in a position to address questions related to explicit computations that one could perform. We have revised this point in the discussion to make it clearer that this relation is conjectural. Making a more quantitative connection between the dynamics on the diagonal induced by the Jacobi algorithm and the dynamics of the eigenvalues in Dyson Brownian motion is an open direction for research.

0.8: Comparison to Ref. [25]. We have included a brief intermediate comment about strong disorder renormalization and its relation to the SJA.

1: The slope of the magenta curve in Fig. 1b matches the black curve at late times. This is coincidence. The same data is plotted for longer times in Fig. 10. While we don't understand the late time behavior of the data well, it is not consistent with either SJA or TDPT. We also note that each of these methods aims to find the full log fidelity curve, not just the slope.

2: Scaling of $\rho_{\mathrm{dec}}$. The definition of $\rho_{\mathrm{dec}}$ is stated in Eq. (15). Equation (16) may be regarded as a normalization condition, though it is a consequence of the definition. Note that $\rho_{\mathrm{dec}}$ is a number density that counts all rotations performed on any state in the Hilbert space, and so naturally grows when the Hilbert space dimension is increased. Perhaps the confusing point is that the natural normalization of $\rho_{\mathrm{dec}}$ is in terms of its second moment, not its integral (zeroth moment). In fact, we expect that the integral of $\rho_{\mathrm{dec}}$ can diverge even at fixed $N$. That is, the Jacobi algorithm might never bring the Hamiltonian to an exactly diagonal form in a finite number of rotations. All that is guaranteed is that the Frobenius norm of the diagonal decreases rapidly.

3: Missing factor of two in Eq. (13). We took the square root of both sides in Eq. (13), which is why the exponent is factor of two smaller.

4: Definition of $\eta$. We are using the same definition as in the previous section. We have included the relevant equation immediately after Eq. (23).

5: Slowly varying assumption. We expect this assumption to be valid whenever we are in the dense regime. This is precisely saying that many rotations are necessary to appreciably change the Hamiltonian, and so any individual rotation typically does little.

6: Cross terms. We now state that we need the average of the cross terms to be small in comparison to the diagonal terms, rather than that they average to zero.

7.:Approximation in Eq. (52). We have corrected the error term in Eq. (52). The next terms are on the order of $w^3 t^2$ and $w^3 \tau^2$. There is an important order of limits where we take w to zero quickly enough in comparison to the increase in t and tau for these terms to be negligible.

8: Error plots in Fig. 7. We maintain that the absolute difference plots in Fig. 8 are meaningful. Note that the Jacobi autocorrelator passes through zero, which makes the relative error poorly behaved. The Jacobi spectral function is positive, but the bare spectral function is extremely small in the tails, which also causes the relative error to be large. This large relative error does not have anything to do with the log-fidelity, which does not depend much on the behavior of the Jacobi spectral function in the large omega tails. It may help to observe that all axes in Fig. 7 are plotted in terms of dimensionless ratios. None of the plots may be freely scaled by a change of units or increasing system size, making the smallness or largeness of the absolute difference compared to 1 a meaningful assessment of the change in the spectral function and autocorrelator as $J/\sigma_\omega$ is increased.

9: Replacing the sum over $\sin^2(\eta/2)$. We have expanded on the discussion in that paragraph. The particular replacement we made was essentially a small angle expansion of $\sin^2(\eta/2)$.

10: Difference between $C_{\mathrm{Jac}}$ and $C_V$. If it is available, one should always use $C_{\mathrm{Jac}}$. $C_V$ may be considered to be the leading order term in an expansion of $C_{\mathrm{Jac}}$ in the perturbation strength $J/\sigma_\omega$. The paragraph Prof. Khaymovich mentions seems to have not been updated from a previous version of the paper. We use $C_{\mathrm{Jac}}$ in both subsections. This has now been corrected.

11: The random matrix model. The perturbing off-diagonal matrix in this model is given by the ETH ansatz for the matrix elements of a local operator in the eigenbasis of a thermalizing Hamiltonian. We are not aware of any in-depth study on the properties of this model from the random matrix perspective, but we have made a comment on some readily apparent features.

12: System sizes in Fig. 8. We have stopped at $N=2^{13}$ as this is already sufficient to observe the asymptotic behavior we seek to describe. There is no fundamental impediment to going to larger sizes, only numerical effort. Also, the many-body numerics are performed in a fixed symmetry sector, so they do not actually use $2^{16}$ states.

13: Missing factor of 2. This has been corrected.

14: Damped oscillations. The invisibility of the damped oscillations in Fig. 8 may be due more to the scale of the plot. We think that the oscillations are visible in panel (b), though they are slightly disguised by the overall decay of the log-fidelity curve. We have included an inset in panel (a) which shows the damped oscillations. One may also explicitly see the oscillatory term in the second line of what is now Eq. (81) in the revised manuscript.

15: Physical origin of discrepancy between SJA and TDPT. The SJA prediction for the log-fidelity may be thought of as a resummation of a particular series in perturbation theory. As such, the physical origin of the difference between SJA and (first order) TDPT would be the occurrence of multiple virtual transitions dressing the eigenstates.

16: Calculating $C_{\mathrm{Jac}}$ needs the eigenvalues. This is not true. The $E_{a_n}$ terms appearing in Eq. (79) are the expectation value of the Hamiltonian in the Jacobi basis at step n. Only when n becomes very large (roughly $O(N^2)$) will this be a good approximation to the actual eigenvalue. Further, $C_{\mathrm{Jac}}$ is not sensitive to this precise value, but only to the density of states (properly, the LDOS). This is much coarser information, and one could in principle obtain it without doing complete diagonalization. It is true that, in running the Jacobi algorithm numerically exactly, we are essentially performing an exact diagonalization calculation to compute $C_{\mathrm{Jac}}$. It is an open avenue of research to predict $C_{\mathrm{Jac}}$ from a microscopic definition of the model, and we are currently pursuing this. We have also clarified these points with an additional paragraph in the discussion. Lastly, we note that, in the random matrix model, the SJA prediction seems to converge to the infinite $N$ limit much faster than the exact calculation. There is thus a significant saving in system size. The complexity of the Jaocbi algorithm as a numerical technique for exact diagonalization is discussed in Sec. 2.1. It is $O(N^3)$, similar to most other simple diagonalization algorithms.

17: Footnote on page 22. The footnote has been brought into the main text.

18: System size. The system size $L$ is a parameter of the model, not a fitting parameter. It cannot be freely chosen so as to optimize the accuracy of the SJA prediction. One must calculate $C_{\mathrm{Jac}}$ for the system size being studied, and then compare to the exact dynamics in that same system size. We expect that the log fidelity density will have a finite thermodynamic limit at fixed $t$, so for large enough $L$ the system size no longer matters. But several of the models we studied are still far from this regime at the system sizes we could simulate.

19: No difference between SJA and TDPT in Fig. 9. Indeed, SJA and TDPT are very similar in Fig. 9. There is, of course, a regime in which TDPT works very well. In this regime, the SJA must necessarily reproduce TDPT in order to be correct. The purpose of Fig. 9 is to show that this is the case. SJA does as well as TDPT when TDPT works well, but also continues to work when TDPT fails (Figs. 8, 10).

---

## Round 4 · Referee Report · Anonymous (Referee 1) · 2023-11-23

Report

I find the authors' replies sufficient and now recommend publication.

---

## Round 4 · Author Response

This revised manuscript includes several clarifications and corrections which address the reports of the referees.

---

## Round 4 · List of Changes

• New comment in the introduction on the relationship between strong-disorder RG and the SJA.
  • New paragraph in Sec. 2.2 on models which show slower than exponential relaxation.
  • Extra column in Table 1 giving the interpretation of each approximation.
  • Corrected error terms in Eqs. (53, 65).
  • New comment on higher order corrections in the perturbation strength in Sec. 4.3.3.
  • New footnote on the estimate of the cutoff time in Sec. 4.3.4.
  • New section on higher order corrections in the inverse many-body bandwidth (Sec. 4.3.5).
  • Corrected description of the numerics at the beginning of Sec. 5.
  • New inset in Fig. 8 showing oscillatory behavior.
  • The footnote in Sec. 5.1 has been brought into the main text, and the damped oscillatory term has been explicitly noted.
  • New paragraph in the discussion on the numerical computation of the Jacobi autocorrelation function and possible future research.
  • Clarified discussion of Dyson Brownian motion in the discussion.
  • New appendix A.4 calculating the leading correction to Eq. (3) in the inverse many-body bandwidth.
  • Minor changes in wording throughout, and additional references.

---

## Editorial Decision

published